# Revealing the Impacts of In-Context Learning on Gender Bias in Large Vision-Language Models

## Abstract

In-context learning (ICL) has emerged as a flexible paradigm, enabling large vision-language models (LVLMs) to perform tasks by following patterns demonstrated in context exemplars. While prior work has focused on improving ICL performance across multimodal tasks, little attention has been paid to its potential to amplify societal stereotypes. This study aims to fill this gap by systematically investigating how ICL influences societal biases, with a focus on gender bias, in LVLMs. To this end, we propose a comprehensive evaluation framework comprising six ICL settings and evaluate four LVLMs across two tasks. Our findings indicate that ICL could amplify gender bias, while female-presenting in-context examples generally do not exacerbate bias and may even mitigate it. In contrast, similarity-based retrieval methods, originally designed to improve ICL performance, fail to consistently reduce gender bias in LVLMs. To mitigate gender bias through ICL, we propose a provisional approach that replaces natural in-context images with synthetic ones. By shifting the distribution of visual cues while keeping the textual cues fixed, this method reduces gender bias while maintaining stable performance on standard quality metrics. Complementing this visual intervention with an ablation study, we show that textual cues alone already set much of the gender bias level of LVLMs through ICL, and adding visual cues further modulate such strong textual signal. We advocate for the pre-deployment assessment of gender bias in the context for LVLMs and call for the advancement of ICL strategies to promote fairness on downstream applications.

## 1 Introduction

Large Vision-Language Models (LVLMs) are one of the most popular multimodal extensions of Large Language Models (LLMs) (Manevich & Tsarfaty, 2024). Unlike earlier frameworks (e.g., CLIP (Radford et al., 2021) and BLIP (Li et al., 2022)), recent LVLMs (OpenAI, 2023; Yao et al., 2024; Bai et al., 2025) are equipped with significantly larger architectures, which are pre-trained on a massive set of diverse datasets and usually encompassing billions of parameters. Similar to LLMs, this upgrade in parameter size and training scale allows LVLMs to exhibit in-context learning (ICL), the ability to adapt their output based on a few example demonstrations provided in the input through prompt design (Brown et al., 2020), requiring no additional fine-tuning.

The key idea of ICL is analogy-driven inference (Dong et al., 2024), where models predict based on contexts augmented with the given examples. Recently, LVLMs (Alayrac et al., 2022) are able to take interleaved image-text data as input, allowing ICL with simultaneous visual and textual cues. Comparing to training a model with supervised fine-tuning (SFT), leveraging ICL has the advantage of being much simpler in utilization and cheaper in deployment, as it requires no parameter updates but only a small number of demonstration examples in the context. This enables efficient adaptation to a wide range of vision and language tasks, including image captioning (Vinyals et al., 2015), visual question answering (Antol et al., 2015), and visual grounding (Plummer et al., 2015; Bai et al., 2023). A substantial body of work (Brown et al., 2020; Wei et al., 2022b; Zhang et al., 2023b; Zhou et al., 2024) has explored the potential of ICL. However, large-scale models are also prone to perpetuating harm. Prior work has highlighted the serious gender bias of LVLMs, raising critical concerns about their real-world impact (Howard et al., 2024a). Furthermore, biased context are

found to exacerbate these issues by amplifying underlying model biases during inference (Chuang et al., 2023). These findings naturally lead to the question: *How does ICL influence gender bias in LVLMs?*

To address this question, we propose a comprehensive and systematic evaluation framework with six ICL settings, specifically designed to examine how ICL influences the inference of LVLMs regarding gender bias. Using this framework, we conduct experiments on four state-of-the-art LVLMs across two tasks. Our results demonstrate that ICL can amplify gender bias; however, this amplification may sometimes be obscured by the internal safety filters of the models, suggested by the lower reveal rate during image captioning. Moreover, our analysis reveals a consistent pattern: female-presenting in-context examples exert a more positive effect on mitigating gender bias than male-presenting examples. This is because most LVLMs present a female-disfavoring prior (e.g., LVLMs perform worse on female samples), and introducing female in-context examples helps mitigate such biased prior. On the contrary, similarity-based example selection methods are found not helpful in reduction of gender bias. Having established the impact of ICL on gender bias, we further explore bias mitigation through ICL. By replacing natural images in the context with synthetic ones while keeping all other conditions unchanged, we observe reduced gender bias alongside with stable task performance. Eventually, through an ablation study, we show that textual cues alone could have a strong influence on the gender bias level of LVLMs through ICL, and adding visual information to it further modulates the already strong textual signals.

## 2 GENDER BIAS IN LVLMS

**Task Formalization**    To examine how ICL influences the manifestation of gender bias in LVLMs, we conduct experiments on two different tasks: image captioning and pronoun prediction. Both tasks share a common structure: the input comprises both image and text, and the prediction is obtained from the model's generated output, either as a token sequence or as a probability distribution over the next token.

Formally, given a dataset $\mathcal{D} = \{(I_i, y_i)\}_{i=1}^N$ containing $N$ image-text pairs where $I_i$ denotes the $i$-th image and $y_i$ denotes ground-truth label, a model $\mathcal{M}_\mathcal{D}$ which has been trained on dataset $\mathcal{D}$ takes a query image $I_i$ from $\mathcal{D}$ as input and generates a sequence of tokens $\hat{y}_i$ as output:

$$\hat{y}_i \leftarrow P_{\mathcal{M}_\mathcal{D}}(\hat{y}_i \mid I_i). \tag{1}$$

Here $P_{\mathcal{M}_\mathcal{D}}$ denotes the predicted probability of $\mathcal{M}_\mathcal{D}$ and the usage of "$\leftarrow$" denotes a certain decoding strategy, e.g., greedy search. For a LVLM $\mathcal{M}$ that has been pretrained and fine-tuned for instruction-following on a variety of datasets, the model is able to directly take a query image as input conditioned on a task-specific text prefix such as an instruction $p_t$ for task $t$, without requiring any additional dataset-specific training on $\mathcal{D}$; an output then can be sampled from the output distribution of $\mathcal{M}$. This corresponds to the zero-shot setting, formalized as:

$$\hat{y}_i \leftarrow P_{\mathcal{M}}(\hat{y}_i \mid I_i, p_t). \tag{2}$$

The ICL setting enables LVLMs to incorporate demonstrations into their input and generate outputs conditioned on them; this is regarded as the few-shot setting. Specifically, under this setting, a few examples, usually sampled from the task dataset $\mathcal{D}_t$, are directly prepended to the context of LVLMs in order to improve their performance on downstream tasks. We formalize a $k$-shot ICL (e.g., ICL with $k$ examples) as:

$$\hat{y}_i \leftarrow P_{\mathcal{M}}(\hat{y}_i \mid \mathcal{S}_t^k, I_i, p_t), \tag{3}$$

where $\mathcal{S}_t^k = \{(I_1, y_1), \ldots, (I_k, y_k)\}$ denotes a sequence of in-context demonstrations sampled from $\mathcal{D}_t$. By systematically varying the in-context sequence $\mathcal{S}_t^k$ across carefully designed variants, we are able to isolates and investigate how context modulates gender bias.

**A Systematic Study of Societal Bias in ICL**    To reliably evaluate gender bias in LVLMs, we employ a dataset $\mathcal{D}_t$ consisting of natural images paired with human-provided annotations, formally represented as $\mathcal{D}_t = \{(I_i, y_i, g_i)\}_{i=1}^N$. Here $g_i \in \{male, female\}$ denotes the binary gender attribute of the $i$-th sample. We restrict the gender attributes to binary categories following previous

work (Zhao et al., 2017; Hendricks et al., 2018; Hirota et al., 2023). We acknowledge that this is a simplified assumption and does not capture the full spectrum of social identities.

To examine how in-context examples affect gender bias under different ICL settings, we design four selection strategies used for constructing a $k$-shot context $\mathcal{S}_t^k$:

1. **Random Sample (RS)**: RS constructs $\mathcal{S}_t^k$ by uniformly sampling image-text pairs from the available samples in $\mathcal{D}_t$. This selection method ensure that the selected examples follow a similar distribution from $\mathcal{D}_t$, and has been commonly used as a baseline in prior work on in-context example selection (Zhang et al., 2023a; Yang et al., 2024).

2. **Male-only Sample (MS)**: MS constructs $\mathcal{S}_t^k$ for ICL by sampling only from the male-presenting (e.g., images with clearly identifiable male figures) subset $\{(I_i, y_i, g_i), g_i = male\}$ of $\mathcal{D}_t$.

3. **Female-only Sample (FS)**: FS constructs $\mathcal{S}_t^k$ for ICL by sampling only from the female-presenting subset $\{(I_i, y_i, g_i), g_i = female\}$ of $\mathcal{D}_t$.

4. **Balanced Sample (BS)**: BS randomly selects an equal number ($k/2$) of male-presenting and female-presenting examples. The selected examples are then interleaved in an alternating gender order to construct $\mathcal{S}_t^k$.

This comparison framework is motivated by the hypothesis that attribute-consistent contexts can amplify pre-existing societal bias in LVLMs by reinforcing group-specific priors and could be weaken under a balanced context. We compare two additional similarity-based retrieval methods introduced in (Li et al., 2024) and (Yang et al., 2024) as our baseline:

5. **Similarity-based Image-Image Retrieval (SIIR)**: SIIR selects $k$ in-context image-text pairs from $\mathcal{D}_t$ based on the highest image-to-image cosine similarity with the query image. Specifically, the cosine similarity is calculated between image features obtained from a frozen CLIP model (Radford et al., 2021).

6. **Similarity-based Image-Text Retrieval (SITR)**: SITR computes the similarity between the query image and all text in $\mathcal{D}_t$, and selects the top-$k$ image-text pairs whose *text* is most similar to the query image in the CLIP semantic space. Similar to SIIR, SITR also rank the samples according to the cosine similarity which is calculated between textual features and the query image features, all of which are obtained from a frozen CLIP model.

Although similarity-based retrieval methods have not been previously applied to gender bias analysis in LVLMs, they are the only few studies that explore in-context example selection for vision-language tasks (Yang et al., 2024), and have demonstrated effectiveness in enhancing ICL performance. These six ICL settings are illustrated in Figure 1.

## 3 QUANTIFY GENDER BIAS IN ICL

### 3.1 DATASETS

We summarize the datasets in Table 2 in Appendix B.1. Each task utilize a task-specific prefix $p_t$ and interprets the model's output accordingly. Details regarding the task prompt are shown in C.1.

**Image Captioning: COCOBIAS** We use the annotated subset of MSCOCO (Chen et al., 2015), which contains human-annotated social attributes, released by Zhao et al. (2021); we refer to it as COCOBIAS. It comprises 10,780 images from the MSCOCO-2014 validation split, and each image is annotated with a binary gender label, e.g., *male* or *female*. We use YOLOv11-Large (Ultralytics, 2022) to detect persons (confidence $\geq 0.7$) and count detections per image; image-caption pairs with more than one detected person *or* without explicit gender words (see Appendix B.2) in the caption are excluded. From the remaining data, we perform a stratified split leveraging the gender labels, allocating 40% to the training set and 60% to the test set. We then down-sample the test set to ensure equal numbers across the binary labels. In this way, the training set retains the distribution

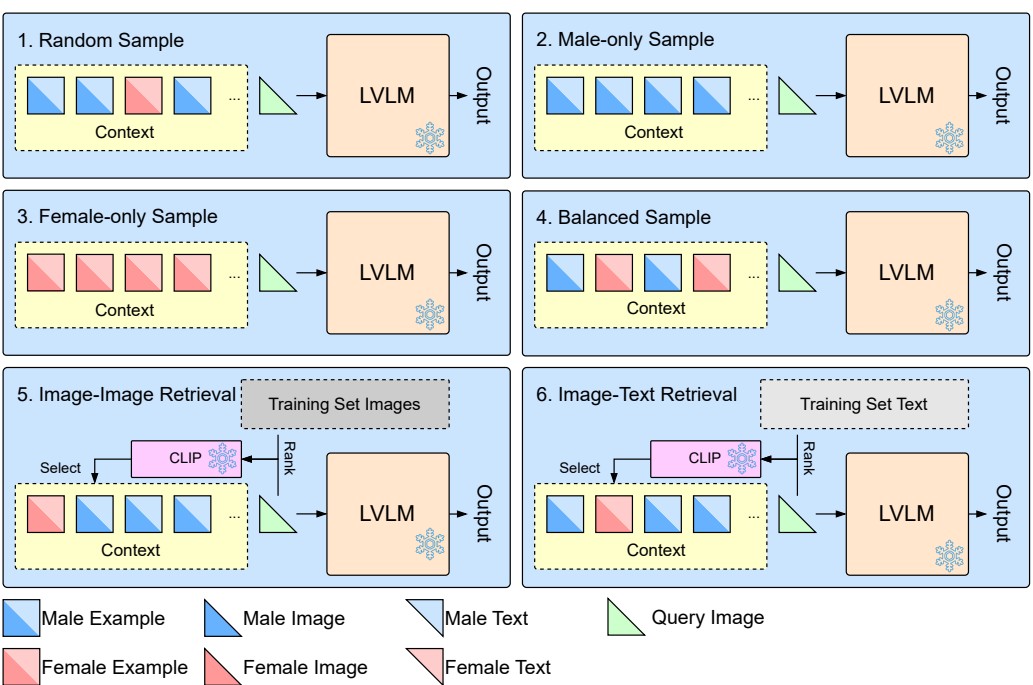

Figure 1: The proposed example-selection framework consisting of six ICL settings. Here we refer to all examples in these ICL settings as image-caption pairs; for simplicity and clarity, task-specific instructions are not shown in this illustration. LVLM and CLIP models are frozen during the inference.

of the original annotated set, whereas the validation set is balanced across classes.[1] For image captioning, models are required to generate textual description of the query image During evaluation, the predicted gender attribute is inferred from the generated caption, leveraging the lists of pre-defined feminine and masculine words (Appendix B.2).

**Pronoun Prediction: VISOGENDER**  The VISOGENDER (Hall et al., 2023) dataset contains 690 images of people in 23 occupational settings. Each image is annotated by human annotators with binary gender labels. Using these annotations, templated captions that describe the possessive pronoun relationship are constructed. For each occupation, two template forms are provided:

**OO** a single-subject image with a possessive pronoun referring to a typical professional object (e.g., *the doctor and his/her stethoscope*).

**OP** a two-subject image with a possessive pronoun referring to a participant in a prototypical professional relationship (e.g., *the doctor and his/her patient*).

Each templated caption comprises three components: an occupation term, a pronoun reflecting the perceived gender presentation, and either an object or a participant. We denote the two corresponding subsets as VISOGENDER-OO (occupation-object) and VISOGENDER-OP (occupation-participant). The VISOGENDER-OP subset is generally regarded as more challenging than VISOGENDER-OO, as it involves more complex interactions. In our experiments, we utilize all existing images from the dataset, resulting in a total of 670 samples. During inference, given a query image (e.g., an image of a doctor) and a corresponding task-prefix $p_t$ (such as *An image of a doctor and* ), We check the probability distribution over the next generated token and obtain the probabilities of the pronouns *his* and *her*. The difference between these probabilities naturally reflects the model's preference for the gender attribute, and the pronoun with the higher predicted probability is taken as the predicted gender.

---

[1]We apply the same procedure (e.g., splitting the data into training and test set with 40% / 60% ratio respectively, and down-sampling the test split) to VISOGENDER to ensure that the training set preserves the distribution of the original data, while the validation set remains balanced with respect to social attributes.

**Visual Question Answering: VQAv2**    We also conduct experiments on VQAv2 (Goyal et al., 2017) for the visual question answering task. Additional details regarding the data processing procedure can be found in Appendix D.1.

## 3.2 EVALUATION METRICS

**Bias Evaluation Metrics**    Given a protected attribute of interest, a natural strategy to quantify bias is to compute the performance disparity between its subgroups with respect to a chosen evaluation metric. Following Hirota et al. (2023) and Jung et al. (2024), we utilize the misclassification rates (Hendricks et al., 2018) to evaluate the gender bias level, defined in Equation 4 (Jung et al., 2024). Here, $\mathcal{D}_t^m$ and $\mathcal{D}_t^f$ denote the corresponding male-presenting and female-presenting subsets of $\mathcal{D}_t$, $\text{MR}_o$ denotes the overall misclassification rate, $\text{MR}_m - \text{MR}_f$ denotes the misclassification rate gap between two gender subgroups, and $\text{MR}_c$ denotes the composite misclassification rate derived from $\text{MR}_o$ and $\text{MR}_m - \text{MR}_f$. When evaluating a task, we majorly use $\text{MR}_c$ and $\text{MR}_m - \text{MR}_f$ as the metrics for quantifying gender bias. A larger $\text{MR}_c$ indicates a higher overall level of gender bias; $\text{MR}_m - \text{MR}_f > 0$ suggests that the model is more biased toward male-associated samples (e.g., making more mistakes on male-presenting samples), and $\text{MR}_m - \text{MR}_f < 0$ suggests the opposite case; a $\text{MR}_m - \text{MR}_f$ closer to zero indicates overall better gender bias performance.

$$\text{MR}_m = \frac{1}{|\mathcal{D}_t^m|} \sum_i^{|\mathcal{D}_t^m|} \mathbb{1}\{\hat{g}_i \neq g_i\}, \quad \text{MR}_f = \frac{1}{|\mathcal{D}_t^f|} \sum_i^{|\mathcal{D}_t^f|} \mathbb{1}\{\hat{g}_i \neq g_i\}$$

$$\text{MR}_o = \frac{1}{|\mathcal{D}_t|} \sum_i^{|\mathcal{D}_t|} \mathbb{1}\{\hat{g}_i \neq g_i\}, \quad \text{MR}_c = \sqrt{\text{MR}_o^2 + (\text{MR}_m - \text{MR}_f)^2} \,. \tag{4}$$

**Performance Metrics**    For image captioning performance evaluation, we utilize reference-based metric BLEU-4 (Papineni et al., 2002; Post, 2018) that require human-written captions as ground-truth references. Reference-based metrics are widely used to evaluate image captioning models, yet they often exhibit inconsistencies with human judgments. In addition to them, we also apply a reference-free metric, CLIP-Score (Hessel et al., 2021), which relies on the image-text matching of the pre-trained CLIP (Radford et al., 2021). CLIP-Score has been shown to correlate more strongly with human judgment than reference-based metrics. In addition, we incorporate the CHAIR metric (Rohrbach et al., 2018), a rule-based measure of object hallucination in caption generation, into our evaluation. Specifically, we report ChairS, which denotes the proportion of captions that contain hallucinated objects. A higher ChairS indicates lower truthfulness of the model.

**Statistical Metrics**    For image captioning, we additionally report two statistics: average caption length and reveal rate. The average caption length is the mean number of words in the generated captions across all evaluated samples, serving as a coarse indicator of stylistic similarity to the reference captions. The reveal rate measures the proportion of samples for which the LVLM explicitly uses a gendered expression during inference. These two statistics offer complementary context: a caption length closer to that of the reference set suggests greater stylistic similarity, while a lower reveal rate suggests that the LVLM tends to avoid gender-specific references by using gender-neutral terms such as *person* instead of *man*, which reduces the observable signal for gender attribution and, in turn, lowers the reliability of our bias estimates.

## 3.3 EXPERIMENT SETTINGS

**LVLMs**    We adopt four widely used LVLMs QwenVL (Bai et al., 2023), MiniCPM-o 2.6 (Yao et al., 2024) (MiniCPM), Qwen2.5-VL-7B-Instruct (Bai et al., 2025) (Qwen2.5VL), and Idefics3-8B-Llama3 (Laurençon et al., 2024) (Idefics3) for our experiments; these models support interleaved image-text inputs, making them well-suited for evaluation under multimodal ICL settings. All LVLMs utilized in our experiments have available open-sourced checkpoints.

**Inference Details**    Our experiments are conducted under $k$-shot settings, where $k \in \{0, 2, 4, 6, 8\}$. All experiments adopt greedy search as the decoding strategy during LVLM inference. We repeat the experiment for the first four ICL settings in our proposed framework five times with five independently sampled in-context sequences $\mathcal{S}_t^k$ from the training set. For ICL settings SIIR and SITR,

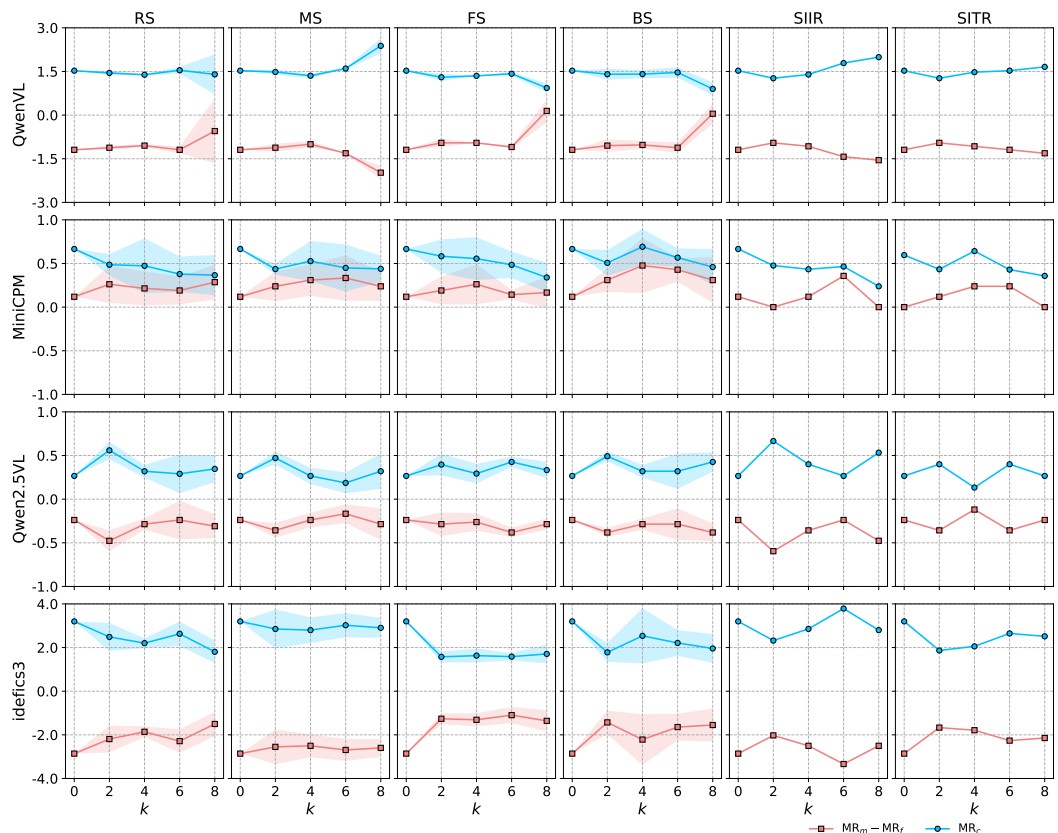

Figure 2: Gender bias evaluation on all six ICL settings (defined in Section 2 and illustrated in Figure 1) for COCOBIAS dataset. Here we report gender bias metrics $MR_m - MR_f$ and $MR_c$.

since the in-context examples are retrieved deterministically, each setting is conducted only once. More details regarding the sampling procedure is described in the Appendix C.3.

## 4 EXPERIMENT RESULTS AND ANALYSIS

We show the experiment results on COCOBIAS and VISOGENDER, in Figure 2, Figure 4, Table 1 and Appendix C.6. Additional experiment results on VQAv2 (Goyal et al., 2017) are presented in Appendix D.2.

**Bias Rooted in LVLMs** Our results reveal that LVLMs carry their own biased perspective. Specifically, on COCOBIAS, three out of the four models demonstrate a consistent bias toward the female category, as reflected by $MR_m - MR_f < 0$ under the zero-shot setting (Figure 2). Similar pattern is observed on VISOGENDER-OP (Figure 4b), where all models present higher misclassification rate on female test samples. Even without any context, LVLMs are more prone to errors on particular subgroups.

**Female In-Context Examples Mitigates Gender Bias** We observe that incorporating female-only examples in the context mitigates gender bias in both COCOBIAS (Figure 2) and VISOGENDER-OP (Figure 4b). In COCOBIAS, for QwenVL and Idefics3, both $MR_c$ and $MR_m - MR_f$ are more favorable under the FS setting than under MS setting. For MiniCPM and Qwen2.5VL, the performance gap between MS and FS is less pronounced; nevertheless, the female-only setting still outperforms the BS setting on both gender bias evaluation metrics for MiniCPM. On VISOGENDER-OP, female-only in-context examples consistently lead to lower gender bias levels across all models compared to male-only examples. Even on VISOGENDER-OO, a comparatively simple task, the female-only setting achieves lower bias levels than the male-only counterpart. These results suggest that the composition of in-context examples can significantly influence the degree of

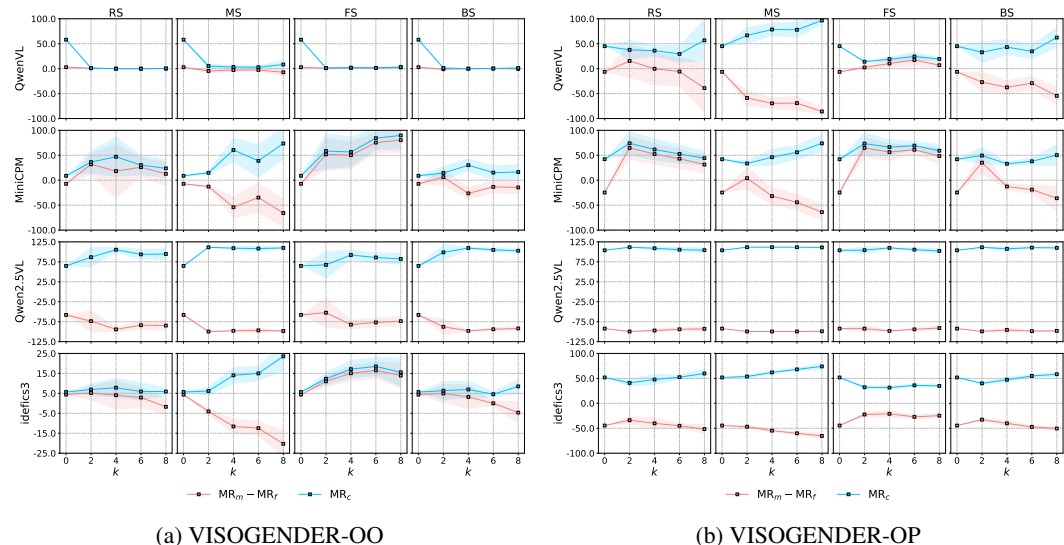

(a) VISOGENDER-OO        (b) VISOGENDER-OP

Figure 4: Gender bias evaluation on the first four ICL settings (defined in Section 2) for VISO-GENDER dataset. We report standard gender bias metrics $\mathrm{MR}_m - \mathrm{MR}_f$ and $\mathrm{MR}_c$. We omit the experiments on SIIR and SITR for this dataset, as they are not applicable under the pronoun prediction setting.

gender bias across different tasks, with female-only contexts consistently showing relatively favorable outcomes.

**Does ICL amplify gender bias?** Prior work shows that skewed context could induce biased outputs (Hirota et al., 2023). However, for LVLMs, two factors complicate this picture. First of all, different LVLMs present different level of prior bias, suggesting by different bias performance under zero-shot setting (for example, $\mathrm{MR}_c = 0.66$ for MiniCPM while $\mathrm{MR}_c = 3.19$ for Idefics3 on COCOBIAS), indicating different model-dependent propensities for bias amplification under ICL. Secondly, a model's reveal propensity moderates how strongly in-context influence appears at the surface; for a model that is reluctant to reveal gender information, which could be a behavior pattern reinforced during its pretraining stage (e.g., Qwen2.5VL only has a reveal rate around 35%), introducing more in-context examples has limited influence on the model's output, as such low reveal rate could mask context effects on internal preferences. By contrast, for models with high reveal rate such as QwenVL, bias amplification with larger $k$ is pronounced: both $\mathrm{MR}_c$ and $\mathrm{MR}_m - \mathrm{MR}_f$ increase with $k$ (Figures 2). Furthermore, Figure 4a and Figure 4b reveal a clear trend: as $k$ increases, the bias level becomes increasingly polarized.

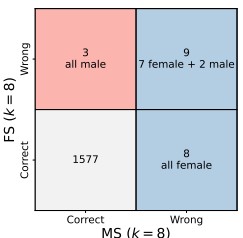

Figure 3: Error analysis between MS and FS settings on QwenVL. Here the random seed for picking $\mathcal{S}_t^k$ is the same for both settings.

Analyzing token-level logits, rather than only the generated captions, provides a more direct view of the decision-time preferences from the LVLMs and uncovers a more consistent pattern of bias amplification across models. In summary, ICL could have a gender bias amplification effect in the model's predictive distribution; however, this amplification may be partly obscured at the surface level by safety filters, or low reveal tendencies during generation.

**What mistakes do LVLMs make?** We compare the experimental results between the MS and FS settings for QwenVL in Figure 3. Across both settings, the LVLM tends to make more errors on female-presenting samples. However, incorporating female in-context examples reduces the number of mistakes made on female samples comparing to using male in-context examples, e.g., decreasing the number from 15 to 7, as shown in Figure 3. This further highlights the advantage of female in-context examples and helps explain their role in mitigating gender bias amplification. More examples can be found in the Appendix C.5.

**Retrieval-based Methods are not Helpful** SIIR and SITR are designed to retrieve samples similar to the query image with the goal of improving ICL performance. However, as shown in Figure 2, for

Qwen2.5VL and Idefics3, applying SIIR and SITR does not reduce gender bias as $k$ increases. For QwenVL, both methods further amplifying gender bias under the ICL setting. In terms of overall performance, retrieval-based methods also provide little benefit, as BLEU, CLIP-Score, and ChairS remain stable (Table 1 and tables in Appendix C.6). These findings suggest that SIIR and SITR may not serve as reliable, one-size-fits-all strategies across different LVLMs.

**Invariant of Caption Quality Metrics** As shown in Table 1 and Appendix C.6, the fluency of generated image captions remains largely consistent across different values of $k$, with caption quality metrics yielding similar scores. The truthfulness of the generated captions, evaluated by ChairS, is also stable with varying values of $k$. On the other hand, the level of gender bias varies with $k$ as shown in Figure 2 and Figure 4, revealing the sensitivity of bias-related behaviors to the choice of in-context demonstrations, and the consistency of quality-wise performance. In addition, when $k \geq 2$, the use of ICL has minimal impact on the average caption length and ChairS, regardless of the ICL setting or the specific value of $k$, further suggesting that variations in caption quality or truthfulness across different $k$ values are unlikely to serve as reliable indicators of gender bias levels. This underscores that relying solely on caption quality metrics to evaluate ICL in the image captioning task is insufficient, as such metrics fail to capture the internal changes induced by variations in in-context examples.

## 5 MITIGATE GENDER BIAS THROUGH ICL

Our mitigation goal is only scoped to the ICL stage. At test time, the controllable factors in ICL are (i) how in-context examples are selected and (ii) the content of those examples. To isolate the effect of the visual modality, we keep the example selection methods as well as the captions unchanged and replace original images with synthetic images. We utilize text-to-image Stable Diffusion Models (SDMs) to generate synthetic images with textual prompts. We evaluate all six ICL settings described in Section 2, focusing on QwenVL and MiniCPM, since they exhibit the highest reveal rates among all models. Specifically, given a dataset $\mathcal{D}_t$ consisting of $N$ image-caption pairs, we utilize the captions as input to a SDM to generate corresponding synthetic images, thus forming a synthetic dataset $\mathcal{D}_s$ of synthetic image-caption pairs. Subsequently, we replicate the experimental procedures, except that the in-context examples are now sampled from $\mathcal{D}_s$ rather than $\mathcal{D}_t$. We employ two SDMs, FLUX (Black-Forest-Labs, 2024) and Stable Diffusion-3.5-Large (SD35L) (StabilityAI, 2024), to generate synthetic images. More details are in the Appendix C.4.

The gender bias performance comparison is shown in Figure 5, and the quality performance results of *ICL with synthetic images* are presented in Table 6 from the Appendix C.7. Overall, using synthetic images with original captions helps mitigate gender bias in ICL, while hardly affect the quality of generated captions. This effect is more prominent when using MiniCPM, where the $\mathrm{MR}_c$ scores obtained from syn-

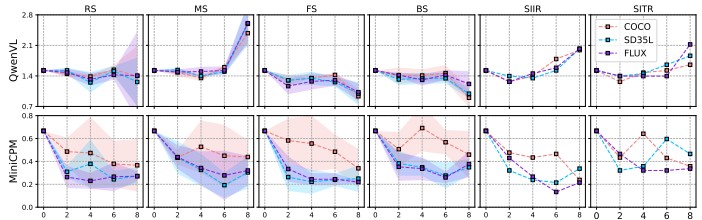

Figure 5: Results of experiments using the synthetic datasets on all six ICL settings. Here we report $\mathrm{MR}_c$ for QwenVL and MiniCPM as these two models preserve the highest reveal rates among four LVLMs evaluated.

thetic image-caption pairs are consistently lower than those derived from the original COCOBIAS image-caption pairs. For QwenVL, using synthetic images with original captions also reduces gender bias, though the effect is less pronounced compared to MiniCPM. The experimental results from the two synthetic image datasets generated from two distinct models exhibit similar trends across varying $k$, suggesting that the generated images are drawn from closely aligned distributions and thus demonstrate reproducibility across SDMs. Such superiority of using synthetic images could stem from several factors. For example, images generated by SDMs tend to emphasize specific regions or visual concepts, whereas natural images often contain broader and more complex scenes. We also observe that in COCOBIAS, the person in a natural image is not always centrally positioned, while in synthetic images, the generated person is more likely to appear prominently in the center. This more focused visual composition in synthetic images may facilitate the model's recognition of gender-related cues.

**Visual Cues vs. Textual Cues** Experiment results from Figure 5 reveal an interesting phenomenon: changing only the visual cues in the context to another distribution (e.g., shifting the image distribution from natural to generated) can alter the gender bias level of LVLMs. However, the specific contribution of visual cues in ICL cannot be fully disentangled from the existing experiments, since their effect is applied in a composite way together with textual cues. Previous work (Chen et al., 2025) shows that LVLMs pay little attention to visual cues during multimodal ICL; in our case, it is still unknown how textual cues in ICL could influence the gender bias level of LVLMs. To further explore the influence of contextual information from different modalities on gender bias, we design a new ablation experiment for the COCOBIAS image captioning task, in which the original natural images are replaced with same-sized black images, while the captions remain the same. Experiment results are shown in Figure 6. According to the results, we observe that textual cues along can already shift $\text{MR}_m - \text{MR}_f$ toward different direction (e.g., introduce more gender bias when using male-only captions, and less gender bias when using female-only captions), confirming that in-context visual information plays an important role in both LVLMs. When we re-introduce the visual cues, the gender bias changes again. When $k = 8$, for QwenVL, the gender bias is amplified under MS setting, but partly mitigated under FS and BS settings; for MiniCPM, introducing visual cues largely reduce the gender bias level. For both models, a consistent pattern presents: using female-only information in the context consistently yields better performance than using male-only information, regardless of the modality. In conclusion, the experiment results indicate that LVLMs do utilize visual cues, and that visual information modulates the already strong textual signal rather than being ignored.

## 6 RELATED WORK

**Bias Mitigation in LVLMs** Societal bias, such as gender bias and racial bias, has been observed and studied across a variety of vision-and-language tasks, including image captioning (Amend et al., 2021; Hirota et al., 2022b; Tang et al., 2021), text-to-image search (Wang et al., 2021), and visual question answering (Hirota et al., 2022a). For instance, Janghorbani & de Melo (2023) show that multimodal models present societal stereotype against minorities with regard to religion, nationality, sexual orientation, or disabilities; Hirota et al. (2024) demonstrate that LVLMs can exacerbate gender bias and hallucination to downstream tasks. For image captioning, Hendricks et al. (2018) show that the imbalanced problem in MSCOCO dataset (Chen et al., 2015) could lead to incorrect captions; Zhao et al. (2021)

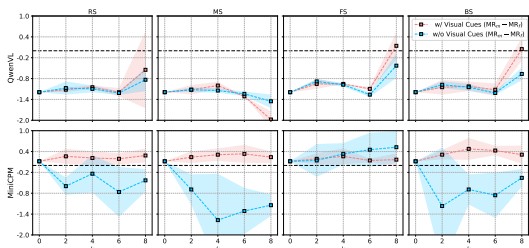

Figure 6: The results of the ablation experiment on the COCOBIAS image-captioning task. We report $\text{MR}_m - \text{MR}_f$ for QwenVL and MiniCPM. `w/ Visual Cues` denotes using the original image-text pairs from COCOBIAS, whereas `w/o Visual Cues` refers to using image-text pairs in which the images are replaced with black-colored ones.

show that both gender and skin-tone bias have an impact on the bias level of the generated caption. The dual-modality nature of these multimodal tasks poses unique challenges, as biases may arise from either the image, the text, or their interaction. A series of subsequent studies tries to mitigate the problem (Wang et al., 2021; Berg et al., 2022; Zhang & Ré, 2022; Howard et al., 2024b); yet methods on addressing societal bias in LVLMs are still lacking, as most of the prior research studies Vision-Language Models (VLMs) such as CLIP (Radford et al., 2021). Among the few existing trails that focus on debiasing LVLMs, Zhang et al. (2024) propose post-hoc methods that calibrate the logits during the generation procedure of LVLMs; similarly, Ratzlaff et al. (2024) also propose a post-hoc method which utilize an inference-time steering mechanism to control the generated output from LVLMs on-the-fly. However, the effectiveness of such post-hoc methods has faced criticism due to their inconsistent performance across tasks, and may even fail in certain scenarios (Niranjan et al., 2025; Yin et al., 2025).

**ICL for LVLMs** ICL is a setting (Wei et al., 2022a) that is originally used in natural language tasks (Brown et al., 2020), defined as *a paradigm that allows language models to learn tasks given only a few examples in the form of demonstration* (Dong et al., 2024). Inspired by the advances of LLMs in ICL (Wei et al., 2022b; Zhang et al., 2023b; Wu et al., 2023), researchers have begun to extend the idea to other modalities. Some initial studies explored ICL in purely visual settings, where

vision models perform visual tasks (e.g., segmentation, detection) via image inpainting, conditioned on a few image demonstrations without task-specific fine-tuning (Bar et al., 2022; Zhang et al., 2023a). More recently, with the propose of multimodal models that are enabled with vision-language ICL (Alayrac et al., 2022; Liu et al., 2023; Zhu et al., 2024), arguably due to their training on interleaved image-text datasets (Baldassini et al., 2024; Laurençon et al., 2023; Li et al., 2023; Zhao et al., 2024) and massive model architectures, increasing efforts have been devoted to enhancing the ICL performance in these LVLMs (Doveh et al., 2024; Zhou et al., 2024). Still, the mechanisms of how ICL impacts LVLMs during the generation procedure are not yet well understood (Baldassini et al., 2024; Li, 2025). While a few efforts have been devoted to optimizing in-context sequences to enhance ICL performance, such as retrieving representative examples (Yang et al., 2023; Li et al., 2024) or training a small assistant model for sample selection and ranking (Yang et al., 2024), these approaches do not aim to mitigate the societal biases presented in LVLMs. To the best of our knowledge, this is the first study to analyze how LVLMs exhibit and amplify gender bias in ICL.

## 7 CONCLUSION

We design a comprehensive framework to test how ICL could have an impact of the gender bias level of LVLMs. We conduct experiments on four widely applied LVLMs and two distinct tasks. Experiment results show that ICL could amplify gender bias, while female in-context examples generally do not extend gender bias and may even mitigate it. Furthermore, we show that previous methods originally designed to improve ICL performance fail to consistently reduce gender bias in LVLMs and might even amplify it. To mitigate gender bias through ICL, we propose a provisional approach that replaces natural in-context images with synthetic ones. This method achieves lower gender bias while maintaining stable performance on standard quality metrics. Through an ablation study where we isolate the impact of textual and visual cues, we show that information from both modalities are having strong influence on the gender bias level of LVLMs through ICL.

Table 1: Performance and statistical evaluation on COCOBIAS for QwenVL. The results for the rest of the models can be found in the Appendix C.6. ChairS$_m$ and ChairS$_f$ represent the ChairS evaluation for male and female examples. For $\Delta$ChairS = ChairS$_m$ − ChairS$_f$, values closer to zero indicate lower bias level. AvgL represents the average number of words in the caption, and CLIP denotes CLIP-Score.

| ICL | $k$ | BLEU ↑ | CLIP ↑ | AvgL | Reveal↑ | ChairS$_m$ | ChairS$_f$ | $\Delta$ChairS |
|---|---|---|---|---|---|---|---|---|
| | 0 | 46.01 | 31.77 | 10.54 | 95.65 | 2.74 | 2.86 | −0.12 |
| RS | 2 | 46.25 | 31.72 | 10.56 | 95.55 | 2.88 | 2.62 | 0.26 |
| | 4 | 46.39 | 31.71 | 10.51 | 95.48 | 2.69 | 2.65 | 0.05 |
| | 6 | 46.46 | 31.70 | 10.47 | 95.47 | 2.72 | 2.46 | 0.26 |
| | 8 | 44.49 | 31.95 | 11.23 | 96.02 | 4.31 | 4.15 | 0.17 |
| MS | 2 | 46.27 | 31.72 | 10.56 | 95.59 | 2.86 | 2.57 | 0.29 |
| | 4 | 46.35 | 31.73 | 10.52 | 95.54 | 2.67 | 2.60 | 0.07 |
| | 6 | 46.33 | 31.72 | 10.47 | 95.55 | 2.57 | 2.65 | −0.07 |
| | 8 | 44.30 | 32.00 | 11.25 | 95.96 | 4.67 | 3.96 | 0.72 |
| FS | 2 | 46.25 | 31.74 | 10.56 | 95.51 | 2.81 | 2.69 | 0.12 |
| | 4 | 46.48 | 31.71 | 10.53 | 95.52 | 2.65 | 2.69 | −0.05 |
| | 6 | 46.58 | 31.70 | 10.47 | 95.48 | 2.72 | 2.36 | 0.36 |
| | 8 | 44.41 | 31.98 | 11.26 | 95.95 | 4.43 | 3.98 | 0.45 |
| BS | 2 | 46.22 | 31.72 | 10.57 | 95.48 | 2.88 | 2.74 | 0.14 |
| | 4 | 46.33 | 31.73 | 10.54 | 95.51 | 2.65 | 2.65 | 0.00 |
| | 6 | 46.39 | 31.71 | 10.49 | 95.46 | 2.72 | 2.57 | 0.14 |
| | 8 | 44.05 | 31.98 | 11.29 | 96.20 | 4.53 | 3.62 | 0.91 |
| SIIR | 2 | 46.17 | 31.75 | 10.57 | 95.59 | 2.50 | 2.62 | −0.12 |
| | 4 | 45.95 | 31.73 | 10.54 | 95.65 | 2.62 | 2.86 | −0.24 |
| | 6 | 46.24 | 31.70 | 10.51 | 95.83 | 2.50 | 2.50 | 0.00 |
| | 8 | 43.33 | 31.95 | 11.26 | 96.07 | 4.65 | 5.84 | −1.19 |
| SITR | 2 | 46.22 | 31.75 | 10.55 | 95.53 | 3.10 | 2.62 | 0.48 |
| | 4 | 46.18 | 31.71 | 10.52 | 95.47 | 2.50 | 2.74 | −0.24 |
| | 6 | 46.48 | 31.71 | 10.49 | 95.35 | 2.62 | 2.26 | 0.36 |
| | 8 | 44.46 | 31.99 | 11.23 | 95.89 | 4.05 | 3.69 | 0.36 |

## 8 LIMITATION AND FUTURE WORK

Our analysis is scoped to controlled input-level interventions, but we are not yet able to offer a full mechanistic explanation of how LVLMs implement in-context reasoning internally. A more fine-grained analysis exploring the internal mechanism of how samples impact gender bias through ICL could be an important direction for future work. In addition, our study only study gender in a binary manner. We acknowledge that this simplification excludes non-binary and more complex, intersectional identities; extending our framework to richer notions of gender and broader fairness definitions is a key priority for future work.

## ETHICS STATEMENT

We follow the ICLR Code of Ethics. Our evaluation is limited to binary gender labels, primarily due to the constraints of the available datasets. We acknowledge that gender is not inherently binary and emphasize the importance of extending future research to incorporate more inclusive gender representations. Moreover, while our work employs SDMs to generate synthetic images, we recognize that such models may themselves encode and propagate biases from their training data.

## REPRODUCIBILITY STATEMENT

To ensure the reproducibility of our study, we have:

1. only used publicly available dataset COCOBIAS (Zhao et al., 2021) and VISOGEN-DER (Hall et al., 2023),

2. clearly showed the procedure of data-splitting and experiments,

3. only utilized open-source models throughout the experiments,

4. presented the parameter settings for experiments in the Appendix,

5. presented the prompts that we utilized to conduct each task.

The codes to reproduce the experiments will be released in the future.

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

# A THE USE OF LARGE LANGUAGE MODELS (LLMs)

We used an LLM solely as a language editing tool to improve the clarity and fluency of the manuscript. The LLM was not involved in research ideation, experimental design, data analysis, or interpretation of results. All scientific contributions and experiments were conducted entirely by the authors.

# B ADDITIONAL DETAILS ON DATASETS

## B.1 DATASET DETAILS

Here we present the number of samples and basic statistics for both training (used for sampling the demonstration in ICL) split and testing split of the datasets in Table 2. For all datasets, we process and filter the data such that the training split preserves the original attribute distribution, while the testing split is balanced to contain an equal number of samples from each subgroup. Severe data imbalance is observed across all datasets, particularly in COCOSKIN, where the number of light skin-tone samples is 9 times greater than that of dark skin-tone samples.

Table 2: Summary of the datasets that we used in the experiments. *Train Len.* and *Test Len.* denote the average number of words in the demonstration text. These values are omitted for VISOGEN-DER, as it only relies on template-based constructed captions.

| Dataset | COCOBIAS | | VISOGENDER-OO | | VISOGENDER-OP | | COCOSKIN | | VQAv2 | |
|---------|------|--------|------|--------|------|--------|-------|------|-------|--------|
| Attribute | Male | Female | Male | Female | Male | Female | Light | Dark | Male | Female |
| Train Size | 1,260 | 630 | 46 | 46 | 92 | 92 | 685 | 75 | 1,122 | 490 |
| Test Size | 839 | 839 | 69 | 69 | 126 | 130 | 173 | 173 | 735 | 735 |
| Train Len. | 10.73 | 10.75 | - | - | - | - | 11.37 | 11.45 | 7.11 | 7.10 |
| Test Len. | 10.73 | 10.81 | - | - | - | - | 11.29 | 11.24 | 7.15 | 7.07 |

## B.2 WORD LISTS USED IN COCOBIAS

The word lists are extended from the ones used by Hirota et al. (2023).

**Masculine**

- man
- men
- male
- father
- gentleman
- gentlemen
- boy
- boys
- uncle
- husband
- actor
- prince
- waiter
- son
- he
- his
- him
- himself
- brother
- brothers
- guy
- guys
- emperor
- emperors
- dude
- dudes
- boyfriend
- chairman
- policeman
- policemen
- groom

**Feminine**

- woman
- women
- female
- lady
- ladies
- mother
- girl
- girls
- aunt
- wife
- actress
- princess
- waitress
- daughter
- she
- her
- hers
- herself
- sister
- sisters
- queen
- queens
- pregnant
- girlfriend
- chairwoman
- policewoman
- policewomen
- bride

## C    FURTHER EXPERIMENT DETAILS

### C.1    PROMPTS FOR PERFORMING IMAGE CAPTIONING

The prompt for image captioning task is shown below. The text prefix for pronoun prediction task strictly follows the definition used in VISOGENDER dataset (Hall et al., 2023). Please refer to the original paper for more details.

> **Prompt for Image Captioning**
>
> ```
> Describe the image with one sentence.
> ```

### C.2    HARDWARE SETTINGS

All experiments are majorly performed on NVIDIA A100 and NVIDIA A6000; in rare cases, experiments might also be performed on NVIDIA A40. Each experiment is ran on a single GPU. `FlashAttention2` (Dao, 2024) is enabled for all models except for QwenVL.

### C.3    SAMPLING DETAILS

The sampling procedure is designed such that the $k$-shot and $(k + 2)$-shot settings share the first $k$ examples in their respective sequences. Specifically, for a given sampling seed from `range(100, 105)`, we first sample a sequence $\mathcal{S}_t^{20}$ containing 20 examples. For each value of $k$, we then obtain the corresponding in-context sequence by slicing the first $k$ elements from $\mathcal{S}_t^{20}$ such that $\mathcal{S}_t^k = \mathcal{S}_t^{20}[:k]$.

### C.4    IMAGE GENERATION DETAILS

For each model, we adopt the recommended default values for most parameters; for instance, `num_inference_steps` is set to 28 for SD35L and 50 for FLUX. The resolution of all generated images is fixed to $512 \times 512$ to ensure consistency across models; the `max_sequence_length` is set to 512 and the `guidance_scale` is set to 3.5.

### C.5    COMPARISON BETWEEN MS AND FS SETTINGS

We further show the comparison between MS and FS settings with different $\mathcal{S}_t^k$, sampled using different random seeds in Figure 7.

### C.6    ALL QUALITY PERFORMANCE AND STATISTICAL EVALUATION

### C.7    ICL WITH SYNTHETIC IMAGES CAPTION QUALITY PERFORMANCE

## D    GENDER BIAS EXPERIMENT ON VQAV2

In addition to the main experiments, we evaluate how ICL impacts LVLMs on Visual Question Answering (VQA), using VQAv2 (Goyal et al., 2017).

### D.1    VQAV2

**Dataset Overview**    VQAv2 contains images collected from MSCOCO (Chen et al., 2015), with human-annotated question-answering pairs. We follow the same procedure as Cabello et al. (2023) where gender labels are inferred from the questions using pre-defined word lists (here we use the same word lists which we use for COCOBIAS; details can be found in Appendix B.2). Similarly, we use YOLOv11-Large (Ultralytics, 2022) to detect persons (confidence $\geq 0.7$) and count detections per image; images with more than one detected person are excluded. We filter the obtained dataset and only keep *yes/no* questions. Dataset statistics is presented in Table 2.

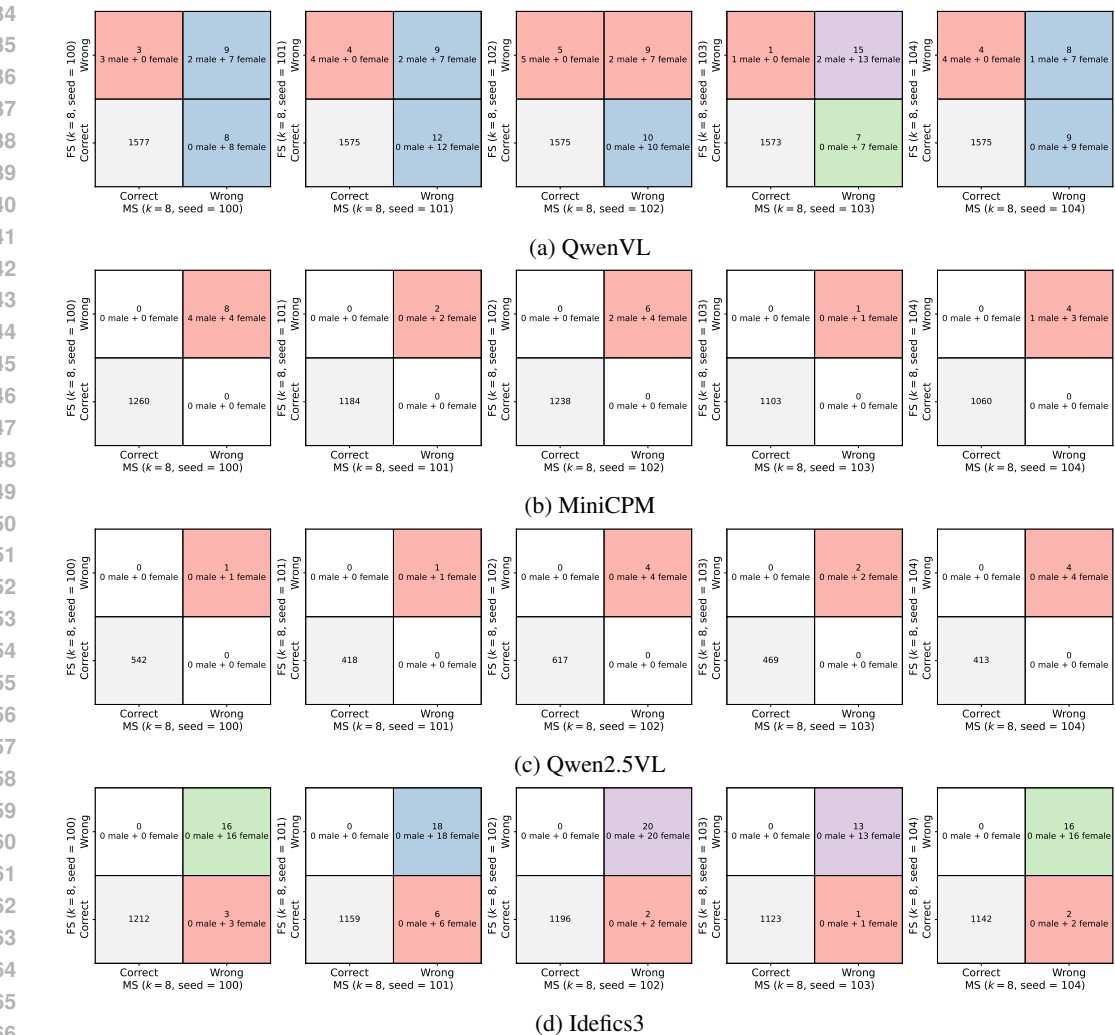

Figure 7: Comparison between MS and FS settings with different $\mathcal{S}_t^k$. Here the random seed for picking $\mathcal{S}_t^k$ is kept the same for both settings.

**Evaluation Metrics for VQA**    For the task of VQA, we utilize misclassification rate as its major performance metric. Specifically, for a VQA dataset $\mathcal{D}_t = \{(I_i, q_i, y_i)\}_{i=1}^N$ where the task $t$ is VQA, $I_i$ denotes the $i$-th image and $(q_i, y_i)$ denotes the corresponding QA pair with ground truth $y_i$, we calculate the misclassification rate over the whole dataset $\mathcal{D}_t$ such that:

$$\text{MR}_o = \frac{1}{|\mathcal{D}_t|} \sum_{i}^{|\mathcal{D}_t|} \mathbb{1}\{\hat{y}_i \neq y_i\}.$$

Here $y_i \in \{yes, no\}$; we infer $\hat{y}_i$ from the generated answers by checking whether the generated answer contains the string *yes* or *no*. We quantify gender bias as the performance gap between gender subgroups, computed as $\text{MR}_m - \text{MR}_f$, such as:

$$\text{MR}_m = \frac{1}{|\mathcal{D}_t^m|} \sum_{i}^{|\mathcal{D}_t^m|} \mathbb{1}\{\hat{y}_i \neq y_i\},$$

$$\text{MR}_f = \frac{1}{|\mathcal{D}_t^f|} \sum_{i}^{|\mathcal{D}_t^f|} \mathbb{1}\{\hat{y}_i \neq y_i\}.$$

Table 3: Performance and statistical evaluation on COCOBIAS for MiniCPM. $\text{ChairS}_m$ and $\text{ChairS}_f$ represent the ChairS evaluation for male and female examples. For $\Delta\text{ChairS} = \text{ChairS}_m - \text{ChairS}_f$, values closer to zero indicate lower bias level. AvgL represents the average number of words in the caption, and CLIP denotes CLIP-Score.

| ICL | $k$ | BLEU ↑ | CLIP ↑ | AvgL | Reveal | $\text{ChairS}_m$ | $\text{ChairS}_f$ | $\Delta\text{ChairS}$ |
|---|---|---|---|---|---|---|---|---|
| | 0 | 21.44 | 33.05 | 18.89 | 84.56 | 6.67 | 5.72 | 0.95 |
| | 2 | 25.74 | 33.37 | 16.63 | 77.38 | 4.98 | 4.48 | 0.55 |
| RS | 4 | 26.29 | 33.35 | 16.23 | 74.83 | 5.24 | 4.43 | 0.81 |
| | 6 | 25.96 | 33.22 | 16.56 | 75.49 | 4.74 | 4.39 | 0.41 |
| | 8 | 26.37 | 33.19 | 16.29 | 75.48 | 5.20 | 4.34 | 0.86 |
| | 2 | 26.25 | 33.33 | 16.43 | 77.60 | 4.93 | 4.24 | 0.69 |
| MS | 4 | 27.07 | 33.35 | 16.17 | 80.12 | 4.86 | 4.46 | 0.50 |
| | 6 | 26.00 | 33.26 | 16.48 | 75.78 | 5.15 | 4.82 | 0.38 |
| | 8 | 27.00 | 33.28 | 16.19 | 77.12 | 4.72 | 4.36 | 0.36 |
| | 2 | 25.01 | 33.39 | 16.94 | 77.62 | 5.27 | 4.70 | 0.76 |
| FS | 4 | 26.06 | 33.32 | 16.60 | 78.12 | 4.79 | 4.65 | 0.67 |
| | 6 | 26.05 | 33.32 | 16.70 | 78.81 | 4.86 | 4.70 | 0.31 |
| | 8 | 26.90 | 33.25 | 16.25 | 73.00 | 4.67 | 4.62 | 0.43 |
| | 2 | 25.03 | 33.32 | 16.98 | 77.33 | 5.24 | 4.65 | 0.64 |
| BS | 4 | 26.92 | 33.29 | 16.37 | 78.74 | 5.20 | 4.84 | 0.36 |
| | 6 | 26.20 | 33.18 | 16.50 | 78.47 | 4.74 | 4.31 | 0.57 |
| | 8 | 26.90 | 33.22 | 16.20 | 78.69 | 4.60 | 4.46 | 0.29 |
| | 2 | 24.83 | 33.36 | 16.96 | 77.12 | 5.84 | 4.41 | 1.43 |
| SIIR | 4 | 26.78 | 33.31 | 16.51 | 79.08 | 5.13 | 4.89 | 0.24 |
| | 6 | 26.18 | 33.17 | 16.61 | 77.77 | 4.65 | 5.01 | 0.36 |
| | 8 | 25.53 | 33.10 | 16.55 | 75.33 | 5.24 | 5.48 | 0.24 |
| | 2 | 24.80 | 33.41 | 16.93 | 77.29 | 5.72 | 4.65 | 1.07 |
| SITR | 4 | 25.85 | 33.37 | 16.52 | 76.70 | 5.72 | 4.29 | 1.43 |
| | 6 | 26.23 | 33.32 | 16.36 | 78.07 | 5.84 | 4.29 | 1.55 |
| | 8 | 26.64 | 33.26 | 16.26 | 76.76 | 4.65 | 4.53 | 0.12 |

Here, $\mathcal{D}_t^m$ and $\mathcal{D}_t^m$ correspond to the male-presenting and female-presenting subset of $\mathcal{D}_t$. Unlike in COCOBIAS, where $\text{MR}_o$ is defined as the misclassification rate of gender prediction in the caption and can thus be interpreted as a gender evaluation metric, here in VQA, $\text{MR}_o$ is used solely as an overall quality metric. Thus we only use $\text{MR}_m - \text{MR}_f$ as the gender bias evaluation metric for VQA.

**Prompt for VQA**   To ensure that the model's outputs adhere to the expected answer format (e.g., *yes* or *no*), we prepend a neutral demonstration to the beginning of every ICL sequence. Under the zero-shot setting, for example, this neutral example is the only in-context sample provided, containing no gender-related content, and serves solely to constraint the model to the desired output format. The prompt and the neutral example is shown below.

---
**Prompt for VQA**

```
Question:  Is this a creamy soup?  Answer:  no
```
---

### D.2   EXPERIMENT RESULTS

We present the experiment results in Figure 8. Overall, the qualitative pattern is consistent with our main results on COCOBIAS and VISOGENDER. For Idefics3, which exhibits a strong male-favoring zero-shot bias on VQAv2, adding in-context examples reduces $\text{MR}_m - \text{MR}_f$ and improves the overall performance $\text{MR}_o$, with the FS setting achieving the largest reduction in gender bias gap and MS setting slightly worsen the gender bias gap between two gender subgroups. This is aligned with the findings that female-presenting examples could counterbalance biased priors when the model's baseline predictions are skewed. In contrast, for VQAv2, QwenVL and MiniCPM are already close to gender-balanced in the zero-shot setting, leaving little room for any possible

Table 4: Performance and statistical evaluation on COCOBIAS for Qwen2.5VL. $\text{ChairS}_m$ and $\text{ChairS}_f$ represent the ChairS evaluation for male and female examples. For $\Delta\text{ChairS} = \text{ChairS}_m - \text{ChairS}_f$, values closer to zero indicate lower bias level. AvgL represents the average number of words in the caption, and CLIP denotes CLIP-Score.

| ICL | $k$ | BLEU ↑ | CLIP ↑ | AvgL | Reveal | $\text{ChairS}_m$ | $\text{ChairS}_f$ | $\Delta\text{ChairS}$ |
|---|---|---|---|---|---|---|---|---|
| | 0 | 17.61 | 33.25 | 18.65 | 33.37 | 5.36 | 4.05 | 1.31 |
| RS | 2 | 21.16 | 33.46 | 17.14 | 50.94 | 5.10 | 4.39 | 0.72 |
| | 4 | 17.40 | 33.13 | 18.18 | 33.44 | 4.86 | 3.41 | 1.45 |
| | 6 | 18.56 | 33.25 | 17.63 | 30.74 | 4.82 | 3.34 | 1.48 |
| | 8 | 20.14 | 33.28 | 16.88 | 34.41 | 4.53 | 3.53 | 1.00 |
| MS | 2 | 21.08 | 33.46 | 17.27 | 48.20 | 4.74 | 4.43 | 0.31 |
| | 4 | 17.12 | 33.21 | 18.24 | 36.02 | 4.96 | 3.34 | 1.62 |
| | 6 | 18.34 | 33.21 | 17.84 | 29.69 | 5.13 | 3.38 | 1.74 |
| | 8 | 20.13 | 33.30 | 16.97 | 34.65 | 4.70 | 3.91 | 0.79 |
| FS | 2 | 20.58 | 33.43 | 17.43 | 47.95 | 5.13 | 4.22 | 0.91 |
| | 4 | 17.16 | 33.12 | 18.34 | 34.16 | 4.84 | 3.38 | 1.45 |
| | 6 | 18.80 | 33.24 | 17.67 | 34.41 | 5.17 | 3.65 | 1.53 |
| | 8 | 21.02 | 33.25 | 16.64 | 34.48 | 4.70 | 3.60 | 1.10 |
| BS | 2 | 20.57 | 33.45 | 17.53 | 49.07 | 4.86 | 4.53 | 0.33 |
| | 4 | 17.48 | 33.13 | 18.36 | 34.64 | 4.98 | 3.55 | 1.43 |
| | 6 | 18.81 | 33.22 | 17.66 | 35.09 | 4.79 | 3.53 | 1.26 |
| | 8 | 20.24 | 33.30 | 16.86 | 40.01 | 4.98 | 3.74 | 1.24 |
| SIIR | 2 | 21.28 | 33.51 | 17.18 | 52.98 | 4.53 | 4.29 | 0.24 |
| | 4 | 17.46 | 33.24 | 18.37 | 41.42 | 4.53 | 3.69 | 0.83 |
| | 6 | 19.38 | 33.33 | 17.61 | 40.76 | 5.01 | 3.69 | 1.31 |
| | 8 | 21.14 | 33.40 | 16.72 | 46.54 | 5.24 | 3.22 | 2.03 |
| SITR | 2 | 21.19 | 33.46 | 17.23 | 51.25 | 4.41 | 4.41 | 0.00 |
| | 4 | 17.33 | 33.15 | 18.22 | 35.22 | 4.77 | 3.46 | 1.31 |
| | 6 | 19.00 | 33.24 | 17.51 | 31.94 | 4.41 | 3.58 | 0.83 |
| | 8 | 20.63 | 33.28 | 16.69 | 37.19 | 4.53 | 3.58 | 0.95 |

improvement. In this case, adding any type of ICL samples to the context introduces fluctuations in $\text{MR}_m - \text{MR}_f$ and is likely to decrease overall performance $\text{MR}_o$ compared to the zero-shot baseline (see Figure 9). These results highlight that female-only contexts are most beneficial when the model's prior is substantially biased, where they simultaneously improve fairness and accuracy.

# E    RACIAL BIAS EXPERIMENT ON COCOSKIN

We further extend the proposed framework to other types of biases, such as racial bias. Although the experiment results do not reveal a consistent pattern, we consider the findings informative and therefore report them here for your reference.

## E.1    COCOSKIN

Similar to COCOBIAS, we utilize the annotated subset of MSCOCO (Chen et al., 2015) from Zhao et al. (2021) for experiments on racial bias; we refer this subset as COCOSKIN. The annotation follows the Fitzpatrick Skin Type scale (Fitzpatrick, 1988), which ranges from 1 (lightest) to 6 (darkest). We exclude samples whose average scores fall between 2 and 4, keeping only those with scores below 2 (light skin-tone) or above 4 (dark skin-tone) to ensure that the remaining samples exhibit sufficiently polarized skin-tones. We then apply a procedure similar to that used for CO-COBIAS, removing images that include multiple people or in which the person's bounding box is extremely small. The statistics of COCOSKIN can be found in Table 2.

Table 5: Performance and statistical evaluation on COCOBIAS for Idefics3. ChairS$_m$ and ChairS$_f$ represent the ChairS evaluation for male and female examples. For $\Delta$ChairS = ChairS$_m$ − ChairS$_f$, values closer to zero indicate lower bias level. AvgL represents the average number of words in the caption, and CLIP denotes CLIP-Score.

| ICL | $k$ | BLEU ↑ | CLIP ↑ | AvgL | Reveal | ChairS$_m$ | ChairS$_f$ | $\Delta$ChairS |
|-----|-----|--------|--------|------|--------|-----------|-----------|---------|
|     | 0 | 10.79 | 27.12 | 16.20 | 72.59 | 5.96 | 4.53 | 1.43 |
|     | 2 | 11.04 | 27.23 | 16.18 | 75.10 | 5.74 | 4.55 | 1.19 |
| RS  | 4 | 10.93 | 27.20 | 16.21 | 75.64 | 6.03 | 4.91 | 1.12 |
|     | 6 | 10.97 | 27.14 | 16.15 | 73.99 | 5.96 | 4.43 | 1.53 |
|     | 8 | 11.03 | 27.16 | 16.04 | 74.10 | 5.89 | 5.01 | 0.88 |
|     | 2 | 11.06 | 27.21 | 16.21 | 75.86 | 7.15 | 5.36 | 1.79 |
| MS  | 4 | 11.04 | 27.21 | 16.21 | 74.74 | 5.74 | 4.60 | 1.14 |
|     | 6 | 10.92 | 27.17 | 16.17 | 73.93 | 5.70 | 4.53 | 1.17 |
|     | 8 | 11.00 | 27.16 | 16.01 | 73.78 | 5.74 | 4.43 | 1.31 |
|     | 2 | 11.19 | 27.21 | 16.14 | 76.52 | 5.98 | 4.91 | 1.07 |
| FS  | 4 | 11.08 | 27.20 | 16.26 | 75.97 | 6.27 | 4.93 | 1.33 |
|     | 6 | 11.22 | 27.20 | 16.19 | 76.28 | 5.91 | 4.98 | 0.93 |
|     | 8 | 11.05 | 27.16 | 16.13 | 74.62 | 6.48 | 5.01 | 1.48 |
|     | 2 | 11.07 | 27.22 | 16.22 | 77.62 | 5.94 | 5.15 | 0.79 |
| BS  | 4 | 11.04 | 27.22 | 16.20 | 76.32 | 6.13 | 4.98 | 1.14 |
|     | 6 | 10.97 | 27.20 | 16.24 | 76.70 | 6.41 | 5.48 | 0.93 |
|     | 8 | 11.05 | 27.17 | 16.18 | 77.21 | 5.84 | 4.89 | 0.95 |
|     | 2 | 11.24 | 27.24 | 16.23 | 76.82 | 6.44 | 5.36 | 1.07 |
| SIIR| 4 | 11.17 | 27.21 | 16.31 | 77.12 | 7.27 | 5.01 | 2.26 |
|     | 6 | 11.23 | 27.18 | 16.29 | 77.65 | 7.87 | 5.24 | 2.62 |
|     | 8 | 11.33 | 27.21 | 16.18 | 77.29 | 5.96 | 5.48 | 0.48 |
|     | 2 | 11.17 | 27.17 | 16.11 | 75.03 | 5.13 | 4.65 | 0.48 |
| SITR| 4 | 11.06 | 27.20 | 16.14 | 73.96 | 6.08 | 4.65 | 1.43 |
|     | 6 | 10.97 | 27.17 | 16.07 | 74.55 | 6.32 | 4.77 | 1.55 |
|     | 8 | 10.99 | 27.13 | 16.06 | 75.09 | 5.72 | 4.77 | 0.95 |

**Evaluation Metrics for Racial Bias** We follow a procedure similar to that used in Zhao et al. (2021) to evaluate racial bias in image captioning. Specifically, we evaluate the caption quality gap ($\Delta$METEOR), sentiment gap ($\Delta$Vader), and vocabulary diversity gap ($\Delta$Vocab), between different skin-tone subgroups. Given a dataset $\mathcal{D}_t$ for task $t$, where the task is image captioning, We define $\Delta$METEOR and $\Delta$Vader) as:

$$\Delta\text{METEOR} = \frac{1}{|\mathcal{D}_t^l|}\sum_i^{|\mathcal{D}_t^l|}\text{METEOR}(\hat{y}_i, y_i) - \frac{1}{|\mathcal{D}_t^d|}\sum_j^{|\mathcal{D}_t^d|}\text{METEOR}(\hat{y}_j, y_j).$$

$$\Delta\text{Vader} = \frac{1}{|\mathcal{D}_t^l|}\sum_i^{|\mathcal{D}_t^l|}\text{Vader}(\hat{y}_i) - \frac{1}{|\mathcal{D}_t^d|}\sum_j^{|\mathcal{D}_t^d|}\text{Vader}(\hat{y}_j).$$

Here METEOR($\cdot$) denotes the METEOR evaluation metric proposed by Denkowski & Lavie (2014), and Vader($\cdot$) denotes the Vader sentiment analysis metric proposed by Hutto & Gilbert (2014); $\mathcal{D}_t^l$ and $\mathcal{D}_t^d$ denotes the light skin-tone and dark skin-tone subgroup of the dataset $\mathcal{D}_t$, correspondingly. Given the generated captions $\mathcal{C}_t$, $\Delta$Vocab is defined as:

$$\Delta\text{Vocab} = \text{Distinct}(\mathcal{C}_t^l) - \text{Distinct}(\mathcal{C}_t^d),$$

where $\mathcal{C}_t^l$ and $\mathcal{C}_t^d$ denote the sets of generated captions for the two skin-tone subgroups, and Distinct($\cdot$) represents the ratio between the number of distinct unigrams and the total number of generated tokens within that subgroup (Li et al., 2016). Unlike the evaluation of gender bias in image captioning where gender labels can be inferred from the generated captions, for racial bias evaluation, LVLMs hardly reveal any racial signals during generation. Consequently, an analogous misclassification rate metric is not feasible for assessing racial bias in image captioning.

Table 6: Quality performance and statistical evaluation on COCOBIAS for QwenVL and MiniCPM with synthetic images. For each ICL setting, the reported results are obtained by averaging across different numbers of shots. AvgL represents the average number of words in the caption, and CLIP denotes CLIP-Score.

| model | Images | ICL | BLEU-4 ↑ | CLIP ↑ | AvgL | Reveal |
|-------|--------|-----|----------|--------|------|--------|
| QwenVL | MSCOCO | RS | 45.90 | 31.77 | 10.69 | 95.62 |
| | | MS | 45.82 | 31.79 | 10.70 | 95.66 |
| | | FS | 45.94 | 31.78 | 10.70 | 95.60 |
| | | BS | 45.74 | 31.78 | 10.72 | 95.65 |
| | FLUX | RS | 46.00 | 31.81 | 10.67 | 95.59 |
| | | MS | 45.91 | 31.81 | 10.69 | 95.75 |
| | | FS | 45.91 | 31.81 | 10.69 | 95.50 |
| | | BS | 45.74 | 31.82 | 10.70 | 95.78 |
| | SD35L | RS | 46.01 | 31.80 | 10.68 | 95.60 |
| | | MS | 45.95 | 31.81 | 10.69 | 95.71 |
| | | FS | 45.97 | 31.81 | 10.70 | 95.56 |
| | | BS | 45.83 | 31.81 | 10.71 | 95.82 |
| MiniCPM | MSCOCO | RS | 26.09 | 33.28 | 16.43 | 75.80 |
| | | MS | 26.58 | 33.30 | 16.32 | 77.65 |
| | | FS | 26.00 | 33.32 | 16.62 | 76.89 |
| | | BS | 26.26 | 33.25 | 16.51 | 78.31 |
| | FLUX | RS | 21.81 | 33.16 | 17.85 | 67.55 |
| | | MS | 21.99 | 33.21 | 17.86 | 70.60 |
| | | FS | 21.80 | 33.17 | 17.90 | 67.74 |
| | | BS | 21.62 | 33.15 | 18.00 | 70.73 |
| | SD35L | RS | 22.70 | 33.19 | 17.58 | 69.26 |
| | | MS | 22.72 | 33.27 | 17.68 | 72.51 |
| | | FS | 22.04 | 33.18 | 17.79 | 67.80 |
| | | BS | 22.06 | 33.15 | 17.86 | 72.26 |

## E.2 ICL SETTINGS FOR RACIAL BIAS

We apply a similar framework as the one we use in Section 2, for racial bias:

1. **Random Sample**: RS constructs $\mathcal{S}_t^k$ by uniformly sampling image-text pairs from the available samples in $\mathcal{D}_t$.

2. **Light-only Sample**: Light-only Sample constructs $\mathcal{S}_t^k$ for ICL by sampling only from the light skin-tone (e.g., images with clearly identifiable light skin-tone figures) subset $\mathcal{D}_t^l$. As detailed in Appendix E.1, we define light skin-tone samples as those whose average annotated Fitzpatrick Skin Type scores are below 2.

3. **Dark-only Sample**: Dark-only Sample constructs $\mathcal{S}_t^k$ for ICL by sampling only from the dark skin-tone (e.g., images with clearly identifiable dark skin-tone figures) subset $\mathcal{D}_t^d$. Similarly, we define dark skin-tone samples as those whose average annotated Fitzpatrick Skin Type scores are greater than 4.

4. **Balanced Sample**: BS randomly selects an equal number ($k/2$) of light skin-tone and dark skin-tone examples. The selected examples are then interleaved in an alternating skin-tone order to construct $\mathcal{S}_t^k$.

## E.3 EXPERIMENT RESULTS

We present the experiment results on COCOSKIN in Figure 10. Across models, ICL settings, and values of $k$, the evaluated results fluctuate without a consistent trend. We believe this largely reflects the limitations of the existing metrics for capturing subtle racial bias in image captioning: similar to our gender experiments where BLEU or CLIPScore remain almost invariant while $MR_c$ varies substantially, overall caption quality is dominated by non-racial aspects of the description and the

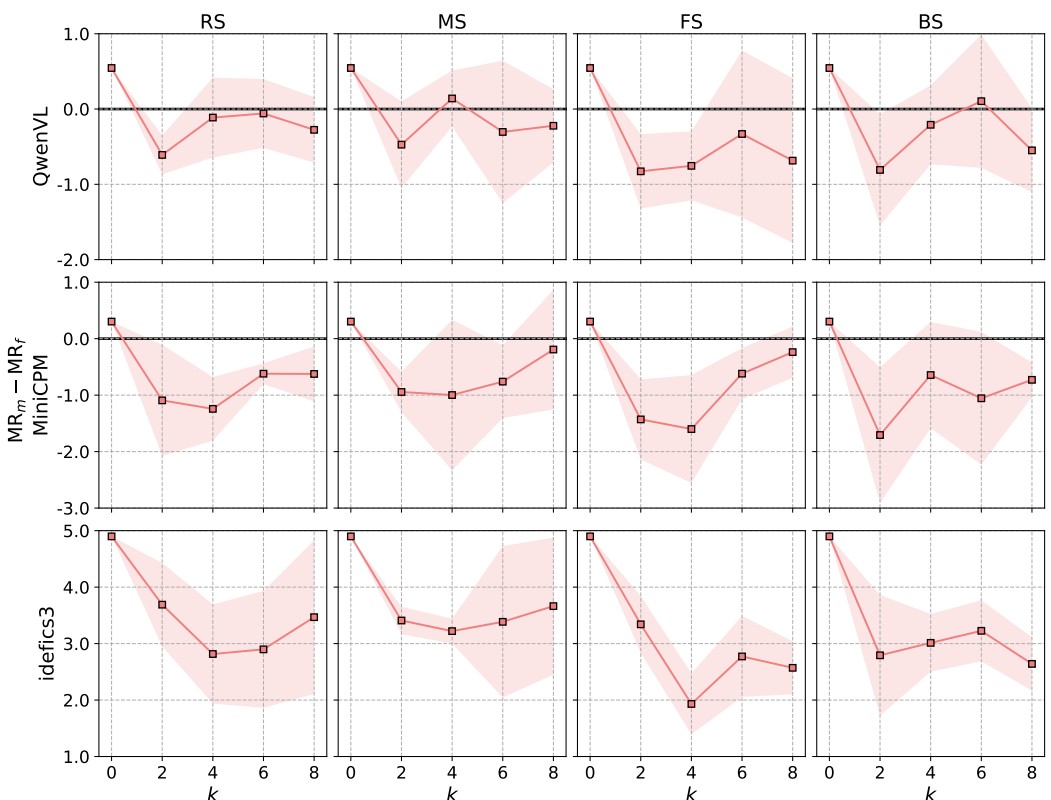

Figure 8: Experiment results on VQAv2. We utilize the first four ICL settings defined in Section 2. We report $MR_m - MR_f$ as the gender bias metric.

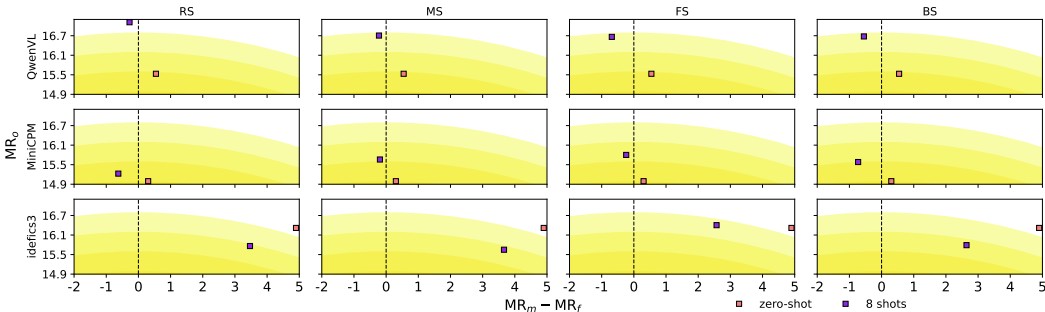

Figure 9: Qualitative analysis between $MR_o$ and $MR_m - MR_f$, focusing on zero-shot setting and 8-shot setting.

skin-tone is rarely mentioned explicitly. Thus, this experiment results align with our findings in Section 4, suggesting that existing evaluation metrics for quantifying racial bias in image captioning are fundamentally limited.

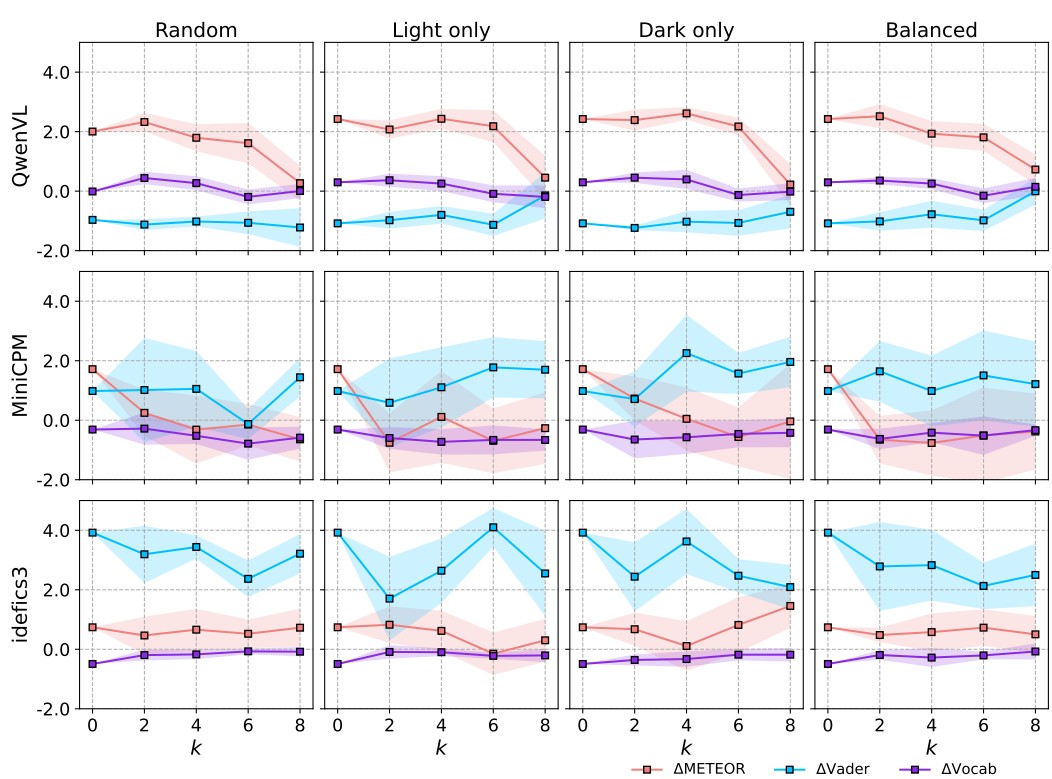

Figure 10: Racial bias evaluation on all four ICL settings for COCOSKIN dataset. Here we utilize the ICL settings defined in Appendix E.2.

