# OpenReview forum: "Revealing the Impacts of In-Context Learning on Gender Bias in Large Vision-Language Models"
_ICLR.cc/2026/Conference — Submitted to ICLR 2026_

### Official Review · Reviewer_FZqq · 2025-10-27

**Soundness:** 3
**Presentation:** 3
**Contribution:** 3
**Rating:** 6
**Confidence:** 4

**Summary:**

This paper studies the ability of in-context examples to affect the gender bias in vision-language model predictions. The authors focus on two tasks: image captioning with the COCOBIAS dataset and pronoun prediction with the VISOGENDER dataset. The study focuses on manipulating the construction of the examples between random, male-only, female-only, balanced, image-image similarity-based, and image-text similarity based. The results with QwenVL, MiniCPM-o 2.6, Qwen2.5-VL-7B, and Idefics-3-8B show that careful construction of the example set can mitigate gender bias, but the error analysis shows that very few of the examples are incorrectly predicted, raising concerns about the nature of this problem in modern VLMs.

**Strengths:**

S1: The paper is clearly written, with a clear structure. The experiments are well-motivated and described. This is an important and interesting problem to study.

**Weaknesses:**

W1: The analyses and supporting figures are too difficult to understand. It takes too much effort for the reader to figure out exactly how "MR_m - MR_f are more favorable under the FS setting than under the MS setting" (L304). Figures 2 and 3 need urgent attention to the y-axis labels, and the captions, reminding the reader about the meaning of RS, MS, FS, Bx, SIIR, and SITR.

W2: The analyses in this paper focus on rather simple next-token completion tasks that are exactly how the models would have been adapted for multimodal ability. The analysis could be improved by considering VQA-style tasks, for which binary gender data splits exist [1].

[1] Cabello et al. EMNLP 2023. Evaluating Bias and Fairness in Gender-Neutral Pretrained Vision-and-Language Models.

**Questions:**

Q1: What is the relationship netween the Bias Evaluation Metrics used in this paper and Bias Amplification?

Q2: Does mutlimodal ICL actually work in these models? The lack of variation in the classic measures in Table 1 suggests that QwenVL might be ignoring the in-context examples.

[2] Wang and Russakovsky. ICML 2021. Directional bias amplification.

---

> ### Author Response · Authors · 2025-11-27
>
> We appreciate the reviewer for the comments and suggestions.
>
> **Regarding the weakness**:
>
> 1. We appreciate the reviewer for the suggestion. We have revised the paper accordingly.
> 2. We thank the reviewer for suggesting adding a new dataset. We add a new dataset to address the concern. Specifically, we add VQAv2 [1] for the task of visual question answering, from which we can infer gender labels from the questions. We follow the same data processing procedure as the one we used for COCOBIAS to ensure that the image for each sample only contains one major human individual. We quantify gender bias as the misclassification rate gap between gender subgroups, denoated as $\text{MR}_m - \text{MR}_f$, and use $\text{MR}_o$ as the overall VQA misclassification rate (Detailed evaluation metric definition can be found in the Appendix D.1). Unlike in COCOBIAS, where $\text{MR}_o$ is defined as the misclassification rate of gender prediction in the caption and can thus be interpreted as a gender evaluation metric, here in VQA, $\text{MR}_o$ can only be used as an overall quality metric. Thus we only use $\text{MR}_m - \text{MR}_f$ as the gender bias evaluation metric for VQA. We present the experiment results in Figure 8. Overall, the qualitative pattern is consistent with our main results on COCOBIAS and VISOGENDER. For Idefics3, which exhibits a strong male-favoring zero-shot bias on VQAv2, adding in-context examples reduces $\text{MR}_m - \text{MR}_f$ and improves the overall performance $\text{MR}_o$, with the FS setting achieving the largest reduction in gender bias gap and MS setting slightly worsen the gender bias gap between two gender groups. This is aligned with the findings that female-presenting examples could counterbalance biased priors when the model’s baseline predictions are skewed. In contrast, for VQAv2, QwenVL and MiniCPM are already close to gender-balanced in the zero-shot setting, leaving little room for any possible improvement. In this case, adding any type of ICL samples to the context  introduces fluctuations in $\text{MR}_m - \text{MR}_f$ and is likely to decrease overall performance $\text{MR}_o$ compared to the zero-shot baseline (see Figure 9 for the qualitative analysis). These results highlight that female-only contexts are most beneficial when the model’s prior is substantially biased, where they simultaneously improve fairness and accuracy. We added the experiment results and corresponding analysis for VQAv2 to the Appendix D.
>
> **Regarding the questions**:
>
> 1. Bias amplification metric [2] focuses on how gender affects task predictions, by comparing correlations in the data versus in the model outputs. Our metric directly compares the downstream performance between gender subgroups. Bias amplification essentially evaluates the word-occurrence to reflect the level of gender bias, while the metrics in our paper use performance gap on downstream tasks as indicator for gender bias. According to [3], in comparison with word-occurrence, evaluating gender bias using downstream metrics could better reflect the gender bias level in real applications.
> 2. We agree that the classic caption quality metrics in Table 1 (BLEU, CLIPScore, ChairS, caption length) vary little with $k$, but we view this precisely as evidence that these metrics are not sensitive to the changes in internal preferences induced by ICL. In Figure 2 and Figure 3, as well as in our new black-image ablation (see our response to reviewer `3tRC`), we observe substantial and systematic changes in $\text{MR}_c$ and $\text{MR}_m - \text{MR}_f$ as $k$ and the composition of in-context examples vary, despite nearly constant quality metrics. We already discussed in Section 4 that ICL mainly reshapes the model’s gender-specific error patterns and reveal rates, while leaving average caption quality almost unchanged. The synthetic-image experiments in Section 5 further reinforce this point: modifying only the visual content of in-context examples significantly reduces gender bias but again has negligible impact on caption quality. Thus, rather than indicating that QwenVL ignores the in-context examples, the stability of Table 1 shows that standard caption metrics can mask important fairness-relevant changes, which is exactly why we need gender bias metrics and token-level analyses.
>
> **Reference**
>
> [1] Goyal, Y., Khot, T., Summers-Stay, D., Batra, D., & Parikh, D. (2017). Making the v in vqa matter: Elevating the role of image understanding in visual question answering. In Proceedings of the IEEE conference on computer vision and pattern recognition (pp. 6904-6913).
>
> [2] Wang, A., & Russakovsky, O. (2021, July). Directional bias amplification. In International Conference on Machine Learning (pp. 10882-10893). PMLR.
>
> [3] Morehouse, K. N., Swaroop, S., & Pan, W. (2025). Rethinking LLM Bias Probing Using Lessons from the Social Sciences. arXiv preprint arXiv:2503.00093.

---

### Official Review · Reviewer_JEjY · 2025-10-28

**Soundness:** 3
**Presentation:** 3
**Contribution:** 3
**Rating:** 6
**Confidence:** 3

**Summary:**

The paper studies whether and how in-context learning (ICL) alters gender bias in large vision-language models (LVLMs). The authors design a six-setting ICL evaluation framework, measure four state-of-the-art LVLMs on image-captioning and VQA, and observe that (i) ICL can indeed amplify gender bias, (ii) placing female-presenting exemplars in the prompt often mitigates bias, (iii) similarity-based example-retrieval does not consistently help, and (iv) synthetic in-context images provide a practical mitigation knob. The work concludes with a recommendation that LVLM deployments should examine the bias of their prompts and that more fairness-aware ICL strategies are needed.

**Strengths:**

Important and timely question: while ICL is now the de-facto way to steer LVLMs, its influence on social bias has been largely unexplored.
Comprehensive evaluation design: six prompt settings, two tasks, four public LVLMs, and both amplification and mitigation analyses.
Interesting empirical findings: (a) ICL occasionally increases gender bias even when overall utility improves; (b) female exemplars are systematically more helpful than male ones; (c) retrieval-based exemplar selection, though popular for accuracy, does not cure bias.
Practical mitigation recipe: swapping natural contextual images for synthetic renders is simple yet shows measurable bias reduction without hurting standard metrics.
Clear call-to-action: the paper argues convincingly that prompt designers should audit bias and that future ICL research must consider fairness.

**Weaknesses:**

Limited definition of “gender bias”: the study focuses on binary male/female stereotypes; it ignores non-binary identities and intersectional dimensions (race, age, etc.).
Causal attribution is still weak: while correlations are shown, the mechanisms of amplification (e.g., attention patterns, token probabilities) are not deeply analysed.
Only two downstream tasks: conclusions may not generalise to other vision-language tasks such as visual reasoning or grounded dialogue.
Dependence on proprietary models: two of the four LVLMs have opaque training data and safety filters, making some observations hard to reproduce or explain.
Synthetic-image mitigation is evaluated with a small set of GAN-generated faces; effectiveness on more complex scenes or in real applications remains unclear.

**Questions:**

Did you control for content drift when replacing natural images with synthetic ones? Could changes in low-level image statistics, not gender attributes, drive the mitigation?
How sensitive are results to the number of in-context exemplars? Does bias amplification scale linearly with more biased examples?
Can your synthetic-image strategy be combined with similarity-based retrieval (i.e., retrieve similar images, then substitute them with synthetic gender-balanced versions)?
Did you examine other social biases (race, age, profession)? If so, are the trends similar; if not, do you foresee any obstacles to extending your framework?

---

> ### Author Response · Authors · 2025-11-27
>
> **[Part 1]**
>
> We thank the reviewer for spending time and effort in providing insights for this paper.
>
> **Regarding the weakness**:
> 1. **Bias definition**: We thank the reviewer for this important observation. We agree that our operationalization of gender is limited to the binary framework male/female, and we acknowledge this as a limitation of the current study. However, within the field of fairness in machine learning, "bias" is not a monolithic concept and can be defined in numerous, often contradictory, ways. Our study intentionally adopts a narrow and concrete definition to enable a focused and measurable investigation. While we recognize the critical need to move beyond the gender binary and incorporate intersectional dimensions, this limitation reflects a broader challenge within the field. However, following the reviewer's suggestion, we extended our experiments by additionally conducting an experiment of racial bias on COCO images with skin-tone annotations, which we call COCOSKIN (more details can be found in Appendix E.1). We construct shot-sampling sets that preserve the original skin-tone distribution while evaluating on a balanced test set (equal numbers of light and dark skin images). We conduct experiments on four ICL settings (detailed definitions are introduced in Appendix E.2) similar to our framework designed for gender bias. We then measure gaps in caption quality between skin groups using METEOR, VADER sentiment, and vocabulary diversity, following same way proposed in [1]. Experiment results are shown in Figure 10 from Appendix E.3, from the revised paper. We observe that, across different models, ICL settings, and values of $k$, these gaps fluctuate without a consistent trend. We believe this largely reflects limitations of these aggregate quality metrics for capturing subtle racial bias: similar to our gender experiments where BLEU or CLIPScore remain almost invariant while $\text{MR}_c$ varies substantially, overall caption quality is dominated by non-racial aspects of the description and the skin-tone is rarely mentioned explicitly (e.g., extremely low reveal rate of explicit racial signals). Thus, this experiment suggests that our ICL configurations do not induce strong race-specific degradation in caption quality, but more targeted racial-bias metrics (e.g., race-specific mismatch rates) are needed to draw conclusions. We have included an discussion regarding this limitation and included all the corresponding details in the Appendix E from the revised paper.
> 2. **Causal attribution**: Our goal in this paper is not to fully solve the mechanistic question of how LVLMs implement in-context reasoning, but to systematically investigate how gender information in the context affects gender bias at inference time. We therefore focus on controlled input-level interventions: varying ICL settings and number of shots under fixed tasks and decoding strategy, analyzing token-level logits in pronoun prediction, and replacing only the images with synthetic counterparts while keeping captions and selection strategies fixed. Across these interventions, gender-bias metrics change in an observable pattern while standard quality metrics remain essentially stable. We have made this scope explicit in the limitation section (Section 8) of the revised paper and point to a more fine-grained mechanistic analysis of internal activations as valuable future work.

---

> ### Author Response · Authors · 2025-11-27
>
> **[Part 2]**
>
> 3. **Downstream tasks**: We thank the reviewer for the suggestion. We add a new dataset and a new downstream task to address the reviewer’s concern. Specifically, we add VQAv2 [2] for the task of visual question answering (VQA), in which we infer gender labels from the questions. We follow the same data processing procedure as the one we used for COCOBIAS to ensure that the image for each sample only contains one major human individual (details can be found in Appendix D.1). The experiment results are shown in Figure 8 from Appendix D2. We quantify gender bias as the misclassification rate gap between gender subgroups, denoated as $\text{MR}_m - \text{MR}_f$, and use $\text{MR}_o$ as the overall VQA misclassification rate (Detailed evaluation metric definition can be found in the Appendix D.1). Unlike in COCOBIAS, where $\text{MR}_o$ is defined as the misclassification rate of gender prediction in the caption and can thus be interpreted as a gender evaluation metric, here in VQA, $\text{MR}_o$ can only be used as an overall quality metric. Thus we only use $\text{MR}_m - \text{MR}_f$ as the gender bias evaluation metric for VQA. Overall, the qualitative pattern is consistent with our main results on COCOBIAS and VISOGENDER (see Figure 8). For Idefics3, which exhibits a strong male-favoring zero-shot bias on VQAv2, adding in-context examples reduces $\text{MR}_m - \text{MR}_f$ and improves the overall performance on $\text{MR}_o$, with the FS setting achieving the largest reduction in gender bias gap and MS setting slightly worsen the gender bias gap between two gender subgroups. This is aligned with the findings that female-presenting examples could counterbalance biased priors when the model’s baseline predictions are skewed. In contrast, for VQAv2, QwenVL and MiniCPM are already close to gender-balanced in the zero-shot setting, leaving little room for any possible improvement. In this case, adding any type of ICL samples to the context  introduces fluctuations in $\text{MR}_m - \text{MR}_f$ and is likely to decrease overall performance $\text{MR}_o$ compared to the zero-shot baseline (see Figure 9). These results highlight that female-only contexts are most beneficial when the model’s prior is substantially biased, where they simultaneously improve fairness and accuracy. We have added the experiment results and corresponding analysis for VQAv2 to the Appendix D.
> 4. **Proprietary models**: We thank the reviewer for raising concerns about the reproducibility of the experiments. None of the LVLMs evaluated in our study relies on a closed or proprietary API; all four models (QwenVL, MiniCPM, Qwen2.5VL, and Idefics3) are released as open-source checkpoints, which enables full local inference and mitigates reproducibility issues, even though some training corpora or safety filter details may remain partially undocumented. Despite differences in model origin and training transparency, we consistently observe the same qualitative patterns across LVLMs, for example, the presence of gender bias in zero-shot settings, and the mitigating effect of female-presenting in-context examples. The cross-model consistency suggests that our findings do not hinge on idiosyncratic or opaque components of any single model, but reflect robust and reproducible behaviors of contemporary LVLMs.
> 5. **GAN-generated faces**: We are confused about this point, as our mitigation in Section 5 does not rely on GAN-generated face images. We use Stable Diffusion (not GANs) and a subset of MSCOCO captions with gender annotations, which contains diverse scenes rather than only faces.

---

> ### Author Response · Authors · 2025-11-27
>
> **[Part 3]**
>
> **Regarding the questions**:
>
> 1. **Content drift**: While we do not explicitly control “content drift” beyond prompting the SDMs with the original captions, we mitigate this variability by repeating all ICL settings over five independent in-context seeds and by using two different SDMs to construct synthetic datasets. The fact that $\text{MR}_c$ consistently decreases, with caption quality remaining stable across seeds and generators, suggests that our findings are robust to the "drift".
> 2. **Low-level image statistics**: Low-level image attributes (e.g., color distribution, background smoothness, centralization of subjects) are strongly correlated with gender, as already documented in previous work [4]. This can influence LVLM predictions, as demonstrated in [3]. However, it remains unclear whether such low-level statistics alone can systematically reduce gender bias in LVLMs. We include this in the future work of the revised paper and it is indeed an interesting direction.
> 3. **Number of in-context exemplars and bias amplification scale**:  According to Figure 2 and 3, overall, we observe that the sensitivity to $k$ is both model and task dependent rather than following a single universal linear relationship. For models with a high reveal rate and stronger prior bias (e.g., QwenVL and Idefics3 on COCOBIAS), male-only or skewed contexts are more likely to increase the gender bias level (e.g., increasing $\text{MR}_c$) as $k$ grows. For models with lower reveal rates or weaker priors (e.g., MiniCPM, Qwen2.5VL), the curves over $k$ are flatter and often saturate after a few examples; additional examples beyond $k = 4$ have diminishing marginal effect on the observable metrics. We additionally analyze the token-level predictive distribution on VISOGENDER-OP: as $k$ increases, the logit/probability gap between “his” and “her” grows in a manner that is approximately linear over the different $k$. This is consistent with our discussion that internal safety filters and low reveal propensity can mask context effects at the surface level, even if the internal logits have been largely influenced by the in-context examples.
> 4. **Synthetic-image strategy combined with similarity-based retrieval**: Yes; we have added the corresponding experiments and updated Figure 5 accordingly.
> 5. **Other social biases**: We additionally perform an experiment of racial bias on COCO images with skin-tone annotations which we call COCOSKIN. We construct shot-sampling sets that preserve the original skin-tone distribution while evaluating on a balanced test set (equal numbers of light and dark skin images). We conduct experiments on four ICL settings similar to our framework designed for gender bias. We then measure gaps in caption quality between skin groups using METEOR, VADER sentiment, and vocabulary diversity, following [1] (details can be found in Appendix E.2). Experiment results are shown in Figure 10. Across models, ICL settings, and values of $k$, these gaps fluctuate without a consistent trend. We believe this largely reflects limitations of these aggregate quality metrics for capturing subtle racial bias: similar to our gender experiments where BLEU or CLIPScore remain almost invariant while $\text{MR}_c$ varies substantially, overall caption quality is dominated by non-racial aspects of the description and the skin-tone is rarely mentioned explicitly (low reveal rate of explicit racial signals). Thus this experiment suggests that our ICL configurations do not induce strong race-specific degradation in caption quality, but more targeted racial-bias metrics (e.g., race-specific mismatch rates) are needed to draw conclusions. We have included a discussion regarding this limitation and included the corresponding analysis in the Appendix E.
>
> **Reference**
>
> [1] Zhao, D., Wang, A., & Russakovsky, O. (2021). Understanding and evaluating racial biases in image captioning. In Proceedings of the IEEE/CVF international conference on computer vision (pp. 14830-14840).
>
> [2] Goyal, Y., Khot, T., Summers-Stay, D., Batra, D., & Parikh, D. (2017). Making the v in vqa matter: Elevating the role of image understanding in visual question answering. In Proceedings of the IEEE conference on computer vision and pattern recognition (pp. 6904-6913).
>
> [3] Hirota, Y., Hachiuma, R., Li, B., Lu, X., Boone, M. R., Ivanovic, B., ... & Yang, C. H. H. (2025). Bias in Gender Bias Benchmarks: How Spurious Features Distort Evaluation. In Proceedings of the IEEE/CVF International Conference on Computer Vision (pp. 8634-8644).
>
> [4] Meister, N., Zhao, D., Wang, A., Ramaswamy, V. V., Fong, R., & Russakovsky, O. (2023). Gender artifacts in visual datasets. In Proceedings of the IEEE/CVF International Conference on Computer Vision (pp. 4837-4848).

---

### Official Review · Reviewer_3tRC · 2025-10-29

**Soundness:** 3
**Presentation:** 3
**Contribution:** 2
**Rating:** 4
**Confidence:** 4

**Summary:**

This paper systematically investigates the impact of in-context learning (ICL) on gender bias in Large Vision-Language Models (LVLMs). The authors propose a comprehensive evaluation framework comprising six ICL settings (Random, Male-only, Female-only, Balanced, and two similarity-based retrieval methods). This framework is used to evaluate four different LVLMs across two tasks: image captioning (on COCOBIAS) and pronoun prediction (on VISOGENDER). The study finds that ICL can amplify existing gender biases, but this effect can be masked by model safety filters or low "reveal rates". A key finding is that female-presenting in-context examples tend to mitigate bias, whereas common similarity-based retrieval methods (designed for performance) fail to consistently reduce it and may even amplify it. As a provisional mitigation strategy, the paper proposes replacing natural in-context images with synthetic ones, which is shown to lower gender bias while maintaining stable task performance.

**Strengths:**

1. Important and Timely Problem: The paper tackles a critical and under-explored area. While ICL for LVLMs is a popular research topic, most work focuses on performance improvements, neglecting the potential for bias amplification. This study provides a much-needed analysis of the fairness implications.
2. Comprehensive Methodology: The proposed framework of six distinct example-selection strategies is a key strength. It allows for a systematic and controlled study of how the composition of in-context examples influences model behavior, moving from random baselines to attribute-specific (MS, FS) and performance-oriented (SIIR, SITR) settings.
3. Thorough Experimentation: The findings are supported by experiments on four modern LVLMs (QwenVL, MiniCPM, Qwen2.5VL, Idefics3) and two distinct tasks (generation and prediction). This demonstrates the robustness and generalizability of the conclusions.

**Weaknesses:**

1. Contradictory Claims on Prior Bias: The paper's explanation for why introducing female examples helps reduce bias relies on the claim that they "help counterbalance the LVLMs’ biased priors against females". This is directly contradicted by the paper's own zero-shot results on COCOBIAS, which found that three of the four models already "demonstrate a consistent bias toward the female category". This inconsistency undermines the core explanation for the paper's main finding.
2. Lack of Mechanistic Explanation: Related to the point above, the paper does a good job of showing what happens (FS mitigates bias) but struggles to provide a deep explanation of why it happens.
3. Limited Scope of ICL Factors: The study focuses exclusively on the composition of the demo examples. However, a large body of work on ICL has shown it to be highly sensitive to other factors, such as the order of the examples (Yang et.al). The "Balanced Sample" setting, for instance, interleaves examples, but it's unknown if the effect would hold if the order were changed (e.g., order the examples with the same label as the query in the beginning or closest to the query).
4. Visual vs. Textual Cues: By claiming ICL can amplify existing gender biases but not so much, this paper assumes that the models are reacting to the visual content of the in-context examples. However, recent research has suggested that multimodal models often pay little attention to visual context in ICL, relying heavily on textual cues instead (Chen et.al). Is it possible that the model is not using the images, and that's why the results show ICL has limited influence?
5. Practicality of Mitigation: The proposed mitigation (using synthetic images)  is interesting but its practical application is unclear. This approach adds a significant computational overhead (running a text-to-image model like SDM) for every set of in-context examples, which may not be feasible for real-time applications. The paper doesn't discuss this trade-off.

Reference

Yang, X., Peng, Y., Ma, H., Xu, S., Zhang, C., Han, Y., & Zhang, H. (2024). Lever LM: configuring in-context sequence to lever large vision language models. Advances in Neural Information Processing Systems, 37, 100341-100368.

Chen, S., Liu, J., Han, Z., Xia, Y., Cremers, D., Torr, P., ... & Gu, J. True Multimodal In-Context Learning Needs Attention to the Visual Context. In Second Conference on Language Modeling.

**Questions:**

See weakness

---

> ### Author Response · Authors · 2025-11-27
>
> **[Part 1]**
>
> We appreciate the reviewer for the valuable comments.
> 1. **Contradictory Claims on Prior Bias**: We thank the reviewer for pointing out the unclear part of the paper. In our metric, *bias toward the female category* means that the model makes more errors on female-presenting samples, effectively disfavoring the female subgroup. On COCOBIAS under the zero-shot setting, three out of four LVLMs exhibit higher misclassification rates for female-presenting images. Our statement that *introducing female in-context examples helps counterbalance the LVLMs’ biased priors against females* was intended to describe the phenomenon in which models initially perform worse on female samples than on male ones, and in which adding female-presenting examples reduces this relative disadvantage. We thank the reviewer again for highlighting the unclear description, and we have revised the paper accordingly.
> 2. **Lack of Mechanistic Explanation**: We appreciate that the reviewer likes our result presentation. Our goal in this paper is not to fully solve the mechanistic question of how LVLMs implement in-context reasoning, but to systematically investigate how gender information in the context affects gender bias at inference time. We therefore focus on controlled input-level interventions: varying ICL settings and number of shots under fixed tasks and decoding strategy, analyzing token-level logits in pronoun prediction, and replacing only the images with synthetic counterparts while keeping captions and selection strategies fixed. Across these interventions, gender-bias metrics change in an observable pattern while standard quality metrics remain essentially stable. We have made this scope explicit in the limitation section (Section 8) and point to a more fine-grained mechanistic analysis of internal activations as valuable future work.
> 3. **Limited Scope of ICL Factors**: We thank the reviewer for the insightful comment. We agree that ICL can be sensitive to the ordering of demonstrations, as shown in [1]. However, our setup follows the same standard ICL protocol which has been utilized in other literatures, including [1], where the query is always placed after the in-context examples, and this position is kept fixed across all conditions. This ensures that differences between settings are not driven by moving the query within the sequence. Our work intentionally focuses on the composition of the in-context examples as the main factor, while keeping other design choices (query position, decoding strategy, etc.,) fixed in order to isolate this effect. As part of our experimental design, the Balanced Sample setting (introduced in Section 2) intentionally adopts an interleaved pattern rather than a random permutation. If we only balanced the global counts but left the order unconstrained, it could happen that most male (or female) examples cluster close to the query, potentially giving one gender a larger weight in the context. This would confound gender composition with influence brought by  sample ordering. By interleaving genders, we keep the local neighborhood around the query approximately balanced, so that we can more cleanly isolate the effect of gender composition itself. In addition, all results are averaged over 5 different sampling seeds, which further mitigates certain order noise.

---

> ### Author Response · Authors · 2025-11-27
>
> **[Part 2]**
>
> 4. **Visual vs. Textual Cue**: We appreciate the reviewer for the valuable comment. This is a very interesting point. We are not assuming that models are only reacting to the visual content of the in-context examples; our major focus is distinguishing how examples in the context could have an impact on the downstream generation.
> Nevertheless, to isolately investigate how the image and text could influence the generation independently, we refer to our experiments in Section 5, which can be viewed as an ablation study in which we compare three image-text pair sets (one natural image set and two generated image sets) that share the same captions but differ in their images. The experiment results in Section 5 show that different models exhibit different levels of resistance to the influence of in-context visual cues; the influence of visual information is smaller for QwenVL but much stronger for MiniCPM, as reflected in the relatively larger ratio of change in their gender bias performance.
> To further show the influence of contextual information from different modalities, we introduce a new ablation experiment for the COCOBIAS image captioning task, in which the original natural images are replaced with same-sized black images, while the captions remain the same. We try this setting on QwenVL and MiniCPM. We present the evaluation results on Figure 6 from Section 5 in the revised version of paper. The blue curves in the Figure 6 ($\texttt{w/o visual cues}$) show that captions alone already shift $\text{MR}_m - \text{MR}_f$ toward different direction (e.g., introduce more gender bias when using male-only captions, and less gender bias when using female-only captions), confirming that textual in-context information plays an very important role in both models. When we re-introduce the visual information ($\texttt{w/ visual cues}$) into the context, the gender bias level changes in a model-dependent way; for example, for QwenVL, when $k=8$, the gender bias is amplified under MS setting, but partly mitigated under FS and BS settings; for MiniCPM, introducing visual information largely reduce the gender bias level. For both models, we observe a consistent pattern: using female-only information in the context consistently yields better performance than using male-only information, regardless of whether the information is in textual form or dual-modality. In conclusion, the experiment results indicate that LVLMs do utilize information from the images, and that visual information modulates the already strong textual signal rather than being ignored. We have included a discussion in the revised paper of how visual and textual cues could impact the gender bias level in Section 5.
> 5. **Practicality of Mitigation**: We agree that online per-query image generation with a SDM model would be impractical. However, our mitigation does not require per-query generation. As described in the paper, we construct one synthetic dataset only once, offline, by using COCOBIAS captions as prompts and generating a single synthetic image per caption. At test time, in-context examples are sampled from this fixed synthetic pool in exactly the same way as when using real images. Thus, the online inference cost of our method introduce no additional budget but is essentially identical to using natural images; the additional cost is a one-time offline preprocessing step, comparable to standard data augmentation.
>
> **Reference**
>
> [1]  Yang, X., Peng, Y., Ma, H., Xu, S., Zhang, C., Han, Y., & Zhang, H. (2024). Lever LM: configuring in-context sequence to lever large vision language models. Advances in Neural Information Processing Systems, 37, 100341-100368.

---

### Author Response · Authors · 2025-12-03

Dear AC and all Reviewers,

We sincerely appreciate the reviewers’ thorough and constructive feedback, and we thank you for the time and care devoted to evaluating our work.

We appreciate all the feedback regarding the contributions of our work. We are encouraged to see that all reviewers recognized the importance and relevance of the problem we study. Several reviewers also highlighted the comprehensiveness of our evaluation framework and experiments, as well as the usefulness of the empirical insights presented in the paper.

During the discussion period, we provided additional experiments addressing the reviewers’ comments and clarified several points that were previously unclear. Specifically, we further conducted:

1. **VQA experiments for gender bias**: by incorporating an additional VQA task into our evaluation, we extend the analysis beyond the two tasks originally in our paper and show that the main findings hold across tasks. This generalizes our findings and strengthens the robustness of both the experiment framework utilized in the paper and also our conclusions about how ICL affects gender bias. (Appendix D)
2. **Image captioning experiments for racial bias**: we further extend our image captioning analysis to social bias beyond gender and evaluate racial bias under ICL. These experiments reveal that existing metrics for assessing racial bias in captioning are fundamentally limited, making reliable measurement difficult. This observed trends align with our main findings, providing additional support for the conclusions of the paper. (Appendix E)
3. **An ablation study exploring the roles of visual and textual cues in the context**: by replacing natural images with black images of the same size within the context, we isolate the contribution of both visual and textual cues in the context. Experiment results show that both visual and textual cues contribute to the LVLMs' behavior under ICL, indicating that LVLMs do not rely on text alone when processing in-context examples. (Section 5)

We believe that these updates respond to the reviewers’ concerns and provide a more comprehensive picture of how ICL influences gender bias in LVLMs, further strengthening the validity of our conclusions.

Once again, we thank all the reviewers and the AC for their thoughtful feedback and the time and effort they devoted to the review process. **The paper has been revised accordingly, with the new content highlighted in orange color.**

---

### Meta-Review · Area_Chair_7hL2 · 2025-12-31

**Summary:**

The paper tackles an important problem and presents a broad empirical study across several LVLMs and settings, but the current submission falls short on (i) interpretability of key results and (ii) providing a coherent explanation for the main effect, especially given patterns that appear inconsistent with the proposed narrative.
The reviewers' initial average score was 5.33. No reviewers responded before the discussion frozen date.

The detailed reasons are as follows:
- As reviewer 3tRC pointed out, I also think that some results in the experiments are contradictory or difficult to explain, thus weakening the overall contribution of the paper. For example, in MiniCPM (Figure 2), the bias is biased towards males (in this paper's definition, the error rate is higher for males), and in this case, FS is better than MS in eliminating the bias. In other models, the bias is biased towards females. However, FS is still better than MS in eliminating the bias. This is difficult to explain and understand. Why is it that, regardless of whether there is male or female bias, FS is better at eliminating the bias? This requires a detailed explanation. The current explanation (introducing female in-context examples helps counterbalance the LVLMs’ biased priors against females) cannot solve this.
- Several reviewers pointed out that the causal/mechanism explanation was quite weak, and this was not adequately addressed during the discussion.
- (Minor) The problem of obscure charts remains unresolved. For example, in Figures 2 and 4, the intuitive bias reduction should be close to 0 (the zero line in the middle). At the very least, the authors should add some auxiliary lines (i.e., use a dash or red line) to indicate whether the bias is decreasing or increasing.

It is hoped that the authors will provide reasonable explanations for some contradictory phenomena during the revision process, and conduct deeper causal/mechanistic research to make the article more solid.

**Reviewer Concerns:**

The reviewers' core concerns lie in contradictory explanations, unclear attributions, and unclear article expression (especially some figures).

- Reviewer JEjY:
Concerned that the paper’s bias definition is narrow (binary gender only) and that causal/mechanistic attribution is weak (effects are shown but not deeply explained), with limited evidence of generalization beyond the evaluated tasks.
​
Also, questions whether the synthetic-image mitigation is robust in realistic settings and whether confounds (e.g., content drift or low-level image statistics) may drive the improvements.​

- Reviewer 3tRC
Finds an apparent inconsistency between the paper’s “priors against females” explanation and the reported zero-shot COCOBIAS results phrased as “bias toward the female category,” which undermines the core narrative.
​
Also argues the paper lacks a deeper mechanistic explanation and does not test key ICL sensitivities, such as demonstration order, while raising the possibility that results are driven more by textual than visual context.​

- Reviewer FZqq
Thinks the analysis and figures are hard to interpret (unclear labels/abbreviations), making it unnecessarily difficult to validate the claims from the plots and metrics.
​
Also questions whether the setup is too “next-token completion”-like and whether multimodal ICL is actually doing much, given limited variation in standard caption quality metrics.

**Reviewer Scores:**

No reviewer replied before the discussion was frozen. Based on the authors' responses, I think the reviewers may keep the same scores.

- Reviewer 3tRC (Original Score 4)
While the rebuttal adds clarifications and some additional experiments, the paper still does not resolve the core inconsistency.​
More broadly, the work remains largely descriptive: it shows that FS often mitigates bias, but does not provide a sufficiently deep or falsifiable mechanism, nor does it test key ICL sensitivities such as demonstration order/position to establish robustness (authors provide some clarification but no experiments).

- Reviewer JEjY (Original Score 6)
The revisions help but do not substantially strengthen causal attribution: the paper still mainly reports correlations between ICL settings and subgroup gaps without a clear mechanism that explains when/why amplification or mitigation occurs.​
In addition, the scope limitations (binary gender focus, limited task coverage) and open questions around the synthetic-image mitigation (potential confounds like content drift or low-level statistics) remain  (authors provide some clarification but no experiments), so the overall contribution level is unchanged.

- Reviewer FZqq (Original Score 6)
Even after the discussion, the presentation issues that make the main results hard to verify remain a major barrier: the key figures/metrics are still not intuitive enough to interpret without significant effort. The question "QwenVL might be ignoring the incontext examples" is not directly answered. The authors primarily addressed the possibility of "insensitivity to metrics," but did not rule out the reviewer's core concern—"the model may not be using context / the ICL signal may be weak."

---

### Decision · Program_Chairs · 2026-01-26

Reject